# On-line single particle analysis of ice particle residuals from mountain-top mixed-phase clouds using laboratory derived particle type assignment

S. Schmidt[1], J. Schneider[1], T. Klimach[1], S. Mertes[2], L.P. Schenk[2], P. Kupiszewski[3,4], J. Curtius[5], and S. Borrmann[1,6]

[1]Particle Chemistry Department, Max Planck Institute for Chemistry, 55128 Mainz, Germany
[2]Leibniz Institute for Tropospheric Research, 04318 Leipzig, Germany
[3]Laboratory of Atmospheric Chemistry, Paul Scherrer Institute, 5232 Villigen, Switzerland
[4]Now at: Department of Meteorology, Stockholm University, 10691 Stockholm, Sweden
[5]Institute for Atmospheric and Environmental Sciences, Goethe-University of Frankfurt am Main, 60438 Frankfurt, Germany
[6]Institute for Atmospheric Physics, Johannes Gutenberg University, 55128 Mainz, Germany

*Correspondence to*: J. Schneider (johannes.schneider@mpic.de)

**Abstract.** In-situ single particle analysis of ice particle residuals (IPR) and out-of-cloud aerosol particles was conducted by means of laser ablation mass spectrometry during the intensive INUIT-JFJ/CLACE campaign at the high alpine research station Jungfraujoch (3580 m a.s.l.) in January-February 2013. During the four week campaign more than 70000 out-of-cloud aerosol particles and 595 IPR were analyzed covering a particle size diameter range from 100 nm to 3 μm. The IPR were sampled during 273 hours while the station was covered by mixed-phase clouds at ambient temperatures between -27 °C and -6 °C. The identification of particle types is based on laboratory studies of different types of biological, mineral and anthropogenic aerosol particles. The outcome of these laboratory studies was characteristic marker peaks for each investigated particle type. These marker peaks were applied to the field data. In the sampled IPR we identified a larger number fraction of primary aerosol particles, like soil dust ($13 \pm 5$ %) and minerals ($11 \pm 5$ %), in comparison to out-of-cloud aerosol particles ($2.4 \pm 0.4$ % and $0.4 \pm 0.1$ %, respectively). Additionally, anthropogenic aerosol particles, such as particles from industrial emissions and lead-containing particles, were found to be more abundant in the IPR than in the out-of-cloud aerosol. In the out-of-cloud aerosol we identified a large fraction of aged particles ($31 \pm 5$ %), including organic material and secondary inorganics, whereas this particle type was much less abundant ($2.7 \pm 1.3$ %) in the IPR. In a selected subset of the data where a direct comparison between out-of-cloud aerosol particles and IPR in air masses with similar origin was possible, a pronounced enhancement of biological particles was found in the IPR.

## 1 Introduction

Depending on their chemical and microphysical properties aerosol particles have a strong impact on the solar radiation budget, an influence on the life-time of clouds, and hence also on precipitation (direct and indirect effect; Lohmann and Feichter, 2005). In the mid-latitudes the formation of precipitation occurs mainly via the ice phase. Ice formation can be initiated in the atmosphere either homogeneously or heterogeneously. Spontaneous freezing of cloud droplets at temperatures lower than -37 °C without any catalysts is called homogeneous

freezing (Cantrell and Heymsfield, 2005). At temperatures > -37 °C only heterogeneous freezing can take place with ice nucleation particles (INP) playing the key role by initiating the freezing process. In mixed-phase clouds supercooled cloud droplets and ice crystals coexist at temperatures between -37 °C and 0 °C. Due to the lower saturation vapor pressure over ice compared to water, under certain thermodynamic conditions ice particles grow at the expense of the supercooled droplets (Wegener-Bergeron-Findeisen process; Findeisen, 1938).

Typically only one out of $10^5$ atmospheric particles has the ability to act as an INP (Rogers et al., 1998; DeMott et al., 2010). The ability of aerosol particles to act as INP depends on the chemical and physical properties, e.g. water insolubility, particle size, existence of an ice active site (Sullivan et al., 2010), as well as the required chemical bonds and crystallographic properties.

Previous laboratory and field studies have suggested that mineral dust (in several types) is one of the most important INP (e.g. DeMott et al., 2003b; DeMott et al., 2003a; Kamphus et al., 2010; Atkinson et al., 2013; Cziczo et al., 2013; Diehl et al., 2014) partially because of its high abundance in the atmosphere (Hoose et al., 2010a). Besides mineral dust organic material of anthropogenic and biological origin is also of particular importance for ice formation (DeMott et al., 2003b; Cziczo et al., 2004b) and also a major component of the atmospheric aerosol in general (Jaenicke, 2005). Meanwhile, biological particles, (e.g. spores, fungi or bacteria) are the most efficient INP at high temperatures (Hoose et al., 2010b). Good ice nucleation ability has also been demonstrated for efflorescent salts (e.g. Abbatt et al., 2006; Wise et al., 2012) and glassy organic material (e.g. Froyd et al., 2010; Murray et al., 2010). Additionally, Tobo et al. (2014) could show that the organic material found in soil dust samples is more important for the ice nucleation ability than the mineral components. Laboratory measurements by Augustin-Bauditz et al. (2016) seem to confirm the findings of Tobo et al. (2014). The ice nucleation ability of soot particles is currently under controversial discussion: Some studies indicated good ice nucleation ability of soot (e.g. Cozic et al., 2008; Pratt et al., 2010; Pratt and Prather, 2010), others found no correlation between IPR/INP and soot (Kamphus et al., 2010; Chou et al., 2013), whereas Cziczo et al. (2013) (for cirrus clouds) and Kupiszewski et al. (2016) (for mixed phase clouds) have shown that black carbon containing particles are depleted in IPR compared to out-of-cloud aerosol. Laboratory experiments have shown a wide spread in the nucleation onset conditions for soot and negative results (no ice nucleation) for some experiments (Hoose and Möhler, 2012).

In order to provide much-needed information on the properties of INP, this study sets out to investigate the chemical composition of IPR in mixed-phase clouds. To achieve this goal, a combination of an ice-selective inlet, the Ice-CVI (Ice Counterflow Virtual Impactor; Mertes et al., 2007), and a single particle mass spectrometer, the ALABAMA (Aircraft-based Laser ABlation Aerosol Mass spectrometer; Brands et al., 2011), was operated at the high alpine Jungfraujoch research site in January-February 2013.

Inspection of the data set showed a high amount of organic aerosol in both IPR and out-of-cloud aerosol particles. In order to better understand the mass spectral signatures and to be able to assign the individual mass spectra to certain particle types, it was necessary to perform an extensive set of laboratory measurements. Different types of typical atmospheric particles such as biological, mineral and organic anthropogenic particles (with sizes roughly between 100 nm and 3 µm), were studied using single particle mass spectrometry (Kamphus et al., 2008; Brands et al., 2011). The aim of these studies was to identify instrument-specific marker peaks for each particle type. Subsequently these results were applied to the Jungfraujoch data set. The chemical

composition of the out-of-cloud aerosol particles is compared to the composition of the sampled IPR and a selected cloud event is compared to an out-of-cloud period having the same air mass origin.

## 2 Measurements and Methods

### 2.1 Aerosol Mass Spectrometer

The size-resolved chemical characterization of the aerosol particles was done with the single particle mass spectrometer ALABAMA (Brands et al., 2011). The ALABAMA consists of three parts: inlet system, detection region and ablation/ionization region. An aerodynamic lens (Liu-type; Liu et al., 1995a, b; Kamphus et al., 2008) and a critical orifice form the inlet system of the ALABAMA, which transmits the particles into the vacuum system and focusses the aerosol particles to a narrow beam. At the exit of the aerodynamic lens the particles are accelerated depending on their particle size to a velocity of about 50 - 100 ms$^{-1}$. For optimal working conditions the critical orifice limits the sampling flow to 80 cm$^3$min$^{-1}$ and reduces the pressure in the aerodynamic lens to 3.8 hPa. The desired lens pressure was set using a critical orifice with a variable diameter to account for the low ambient pressure at the Jungfraujoch (approx. 650 hPa).

A skimmer separates the inlet system and the detection region (second pumping region). The detection region consists of two continuous wave detection lasers (Blu-Ray laser; InGaN, 405 nm), which are orthogonal to the particle beam. The particles pass through the two laser beams and the scattered light is reflected by an elliptical mirror and detected by a photomultiplier tube (PMT). The particle velocity can be determined from the time period a particle needs to pass both detection lasers. By calibration with particles of known size the vacuum aerodynamic particle diameter (DeCarlo et al., 2004) can be determined from the velocity of the particles. Both detection lasers are also used to trigger the ablation laser (pulsed ND-YAG-Laser, 266 nm, 6 – 8 mJ per pulse, 5.2 ns per pulse, max. 21 Hz). If one particle passes both continuous laser beams the electronic control system (designed and built at the Max Planck Institute for Chemistry, Mainz, Germany) sends out a trigger signal to the ablation laser. Subsequently, the pulsed laser fires and vaporizes/ionizes the particle partly or completely, ionizing a fraction of the created gas molecules at the same time. The ions are separated in the Z-shaped bipolar time-of-flight mass spectrometer (TOFWERK AG, Switzerland) by their mass-to-charge ratio (m/z) and finally detected by a microchannel plate (MCP). The ALABAMA measures particles with a vacuum aerodynamic diameter in the size range of 100 nm and 3000 nm. The most efficient detection range is between 200 nm and 900 nm.

Additionally, an optical particle size spectrometer ("Sky-OPC", Grimm, model 1.129, size diameter range (d): d > 0.25 μm, d < 32 μm, calibrated with polystyrene latex (PSL) particles, refractive index = 1.60) connected directly to the ALABAMA inlet system measures the size distribution based on the intensity of the light scattered by the particles.

### 2.2 Single Particle Data Evaluation

The data evaluation was done using the software package CRISP (Concise Retrieval of Information from Single Particles, Klimach, 2012), based on the software IGOR Pro (Version 6, Wave-Metrics) . CRISP includes mass calibration, the conversion of mass spectra into so-called "stick spectra" by integration over the peak width of the ion signals, as well as different possibilities to sort the mass spectra into groups of clusters of similar spectra.

Basically, a clustering algorithm tries to find the optimum number of clusters (i.e. groups of mass spectra) that represent the particle population by their average mass spectrum. By nature of the aerosol particle diversity and the non-uniform ionization in laser ablation ionization, it can't be expected that all particles contained in a cluster equal the average cluster spectrum (Hinz et al., 1999). Rather, each spectrum is assigned to that cluster where the distance metric (in our case one minus Pearson's correlation coefficient $r$) of the single particle spectrum and the averaged cluster spectrum reaches a minimum. The fuzzy c-means algorithm (e.g. Bezdek et al., 1984; Hinz et al., 1999; Huang et al., 2013) differs from the k-means algorithm (Hartigan and Wong, 1979; Rebotier and Prather, 2007) in that way that it accounts for the possibility that one particle may belong to two (or more) clusters by using membership coefficients, whereas the k-means assigns each spectrum strictly to that cluster where correlation with the averaged spectrum is highest.

Here we applied the fuzzy c-means algorithm because sensitivity tests conducted in the framework of a PhD thesis (Roth, 2014) with laboratory-generated particle of known composition and number have shown that the fuzzy c-means better separates the particle types and suffers less from false assignments. Also, various parameters that influence the clustering results were tested by Roth (2014), resulting in a "best choice" that was applied here as well: All mass spectra were normalized to reduce the influence of total signal intensity, and all m/z peaks were taken to the power of 0.5 to reduce the influence of the non-uniform laser ablation ionization, thereby increasing the influence of smaller peaks and decreasing that of larger signals. The "fuzzifier", a weighting exponent used for the calculation of the membership coefficients (Bezdek, 1982; Roth et al., 2016) was set to 1.2. A high number of start clusters was chosen to assure that also rare spectra types are considered in the data evaluation (for instance for the out-of-cloud data set from the JFJ campaign 2013 a number of 200 clusters was chosen; for a known particle composition as sampled during the laboratory studies a number between 10 and 50 clusters was chosen depending on the number of spectra). The start clusters were chosen randomly from the total particle population, under the condition that the correlation coefficient ($r$) between two randomly picked start spectra is less than 0.7. The procedure leading from the clustering algorithm to a certain number of particle types, as illustrated in Fig. 1, was as follows: After mass calibration, the spectra were clustered using the fuzzy c-means algorithm, yielding a certain number of clusters. The resulting cluster number can be lower than the chosen number of start clusters. If this is the case, the number of start clusters was sufficiently high not to suppress rare spectra types. Each cluster includes a certain number ($\geq 1$) of mass spectra based on the calculated membership and distance. From all mass spectra in a cluster an average spectrum is calculated which is used for the identification of the particle type represented by each cluster. All mass spectra which did not fulfill the distance criterion ($1 - r \leq 0.3$) compared to any of the clusters were sorted in the cluster "others". The averaged spectra of each cluster were manually examined with respect to the presence of the marker peaks derived from the reference mass spectra (Section 3.1) and assigned to a certain particle type. The "others" cluster was processed again using the fuzzy c-means algorithm, but with reduced constraints and again the resulting clusters were manually examined and, if possible, assigned to particle types. At the end all clusters of the same particle type were merged, whereas clusters that could not be assigned to a certain particle type were added to the cluster "others".

We report the absolute number of particles of the particle type in a certain time period and the percentage of these particles relative to the total particle population (i.e, the sum of all particle types) measured during the same time period. The uncertainties reported along with these numbers were estimated by manual inspection of a

subset of the data, as described in Roth et al. (2016). The assignment of a certain cluster to a particle type is based on the presence of the reference marker peaks in the averaged cluster mass spectrum. Upon inspection of all mass spectra in one cluster it may occur that the marker peaks (or some of the marker peaks) are not present in an individual mass spectrum. Such a mass spectrum has nevertheless been correctly (from a mathematical point of view) assigned to the cluster by the algorithm, because the overall correlation of the mass spectrum with the cluster average is sufficiently high ($r > 0.7$). This can especially occur in cases when many other peaks are similar, as is often observed for organic particles.

For the error estimation, such particle mass spectra were regarded as "uncertain assignments". The percentage of such uncertainly assigned mass spectra was regarded as the relative error. Of the out-of-cloud data set we inspected two clusters, one assigned to biological particles (338 particles) and one assigned to biomass burning aerosol (473 particles). It turned out that 52 of the 338 inspected "biological" mass spectra (15%), and 48 of the 473 inspected "biomass burning" mass spectra (10%) had to be considered as uncertain. Thus, we conservatively estimated the relative error to be about 15% and generalized this error for the whole out-of-cloud data set.

For the IPR data set, where the absolute number of particles is much lower, it was possible to do a more detailed inspection of the clusters: We inspected one cluster assigned to biological particles, where we found that 28 out of 76 were uncertain (37%), and one cluster of the "PAH/soot" particle type, where 9 out of 23 spectra were uncertain (40%). Those particle types containing only a small number of particles ("industrial metals", "Na + K", "aged material") were completely inspected manually, yielding uncertainties for the "industrial metals" of 14%, of the "Na + K" tape of 0% (no uncertain particles), and of the "aged material" type of 44%. Thus, we estimated the relative error (from uncertain particle type assignment) of the IPR population to be 40% with the exception of the industrial metals (14%) and the "Na + K" type.

These error estimates are conservative upper limits for the error range, because the reference laboratory measurements have shown that, e.g., not all biological particles contain the characteristic marker peaks. It may therefore well be that mass spectra that are similar to the cluster average spectrum of a "biological particle" type are really biological particles, even if they do not contain the marker peaks. Finally, the uncertainty inferred from manual inspection was combined with the Poisson counting statistics error (by error propagation) for each particle type.

Typically, the assignment of mass spectra to a certain particle type relies on the most abundant marker peaks. Therefore, smaller species that are abundant on many or even all particle types might go unnoticed. This is most likely the case for secondary inorganics (sulfate, nitrate) and secondary, oxygenated organics, which may add a coating to mineral dust particles, but the particle signal is still dominated by the mineral dust signatures. Thus it has to kept be kept in mind that a particle type called "mineral" should be read as "mineral-dominated". The only exception we made here is the particle type "lead-containing" where we explicitly state that lead does not represent the whole particle composition. Such a classification of particles is very common in the single particle mass spectrometry literature. The ALABAMA uses a 266 nm ND:YAG laser for particle ablation and ionization, thus we expect the assignment of mass spectra to be similar to other instruments using the same laser wavelength (e.g., ATOFMS, SPASS) of which many results on the abundance of particle types similar to our classification

have been reported (Pastor et al., 2003; Erdmann et al., 2005; Hinz et al., 2006; Pratt et al., 2009; Kamphus et al., 2010; Pratt and Prather, 2010; Sierau et al., 2014).

## 2.3   Laboratory measurements

Classification of the different particle types based on typical marker peaks can be done using published single particle mass spectra and the identified corresponding marker peaks from the single particle mass spectrometer literature. However, dependent on ablation laser wavelength and the energy density at the ablation point, these marker peaks are likely to be instrument-specific. Therefore a large set of laboratory reference mass spectra was recorded using the ALABAMA with the objective to determine instrument-specific marker peaks allowing for a more precise particle type classification. These instrument-specific marker peaks are expected to be valid only for the current configuration of the instruments, because parameters like ablation laser wavelength and energy density are likely to influence the ionization efficiency and the ion fragmentation pattern. Because of the high abundance of organic material in the atmospheric aerosol from natural or anthropogenic emissions (Murphy et al., 2006; Zhang et al., 2007; Kroll and Seinfeld, 2008; Hallquist et al., 2009) the focus was put on the distinction of different types of organic material depending on their sources. Additionally, different mineral particle types were investigated in order to differentiate more unambiguously between biological and mineral aerosol (e.g. soil dust). The laboratory measurements include data recorded at the Max Planck Institute for Chemistry in Mainz and at the AIDA (Aerosol Interactions and Dynamics in the Atmosphere; Möhler et al., 2003; Saathoff et al., 2003) chamber at the Karlsruhe Institute for Technology (KIT).

The various particle types (Table 1) were generated for the measurements as suspension or as supernatant ("washing water", e.g. from pollen or bacteria), as mechanically dispersed solid particles (e.g. cellulose, minerals or ground leaves), or they were directly sampled from the source (e.g. from biomass burning, fuel exhaust, soot, cigarette smoke or cooking/barbeque emissions). No size selection of the generated particles was done before transferring the particles into the ALABAMA. Coating experiments were also conducted with sulfuric acid and secondary organic aerosol (SOA; produced by ozonolysis of α-pinene) coatings on mineral dust particles to mimic atmospheric aging processes. The laboratory reference mass spectra are shown in the supplement (Fig. S1 through S20). It was found that one reference particle type produced several types of spectra which were separated using the clustering algorithm described above. The supplement lists the main clusters with the number of spectra in each cluster.

For the determination of the characteristic marker peaks only those mass spectra that represented the majority (for details see supplement) of the different fragmentation patterns were considered. Using these marker peaks biological, mineral and anthropogenic particle types can be differentiated from each other. However, it has to be taken into account that the same particle type can show different fragmentation patterns and that different particle types can also show similar fragmentation patterns. Thus, for precise identification of the particle type, simultaneous measurements of ions of both polarities (anions and cations) by the mass spectrometer is a great advantage, because in many cases the most characteristics signals are only present in one polarity (predominantly in the cation spectra).

It should be further kept in mind that the detection and ablation/ionization efficiency of the ALABAMA (and of most other single particle instruments) is not equal for all particle types. Additionally, the observation that only a fraction of each reference particle type showed the characteristic marker peaks leads to a further bias of the data. It therefore has to be emphasized that the reported particle numbers and relative abundances refer only to particles detected by the ALABAMA and not to their real abundance in the atmosphere. However, comparison between different particle populations is meaningful, because the same biases hold for all sampling periods.

## 2.4 Field studies in mixed-phase clouds

### 2.4.1 Description of the measurement site

The INUIT-Jungfraujoch campaign took place in January-February 2013 at the High Alpine Research station Jungfraujoch in the Swiss Alps (JFJ, Sphinx Laboratory, 3580 m a.s.l; 7°59'2''E, 46°32'53''N) in the frame work of the DFG (Deutsche Forschungsgemeinschaft)-funded research unit INUIT and the Swiss National Science Foundation-funded project "Interaction of aerosols with clouds and radiation". It was conducted in cooperation with the CLACE-campaign (Cloud and Aerosol Characterization Experiment) which took place at the same time.

Due to the exposed mountain rim position and the altitude of the Jungfraujoch the Sphinx Laboratory is mainly situated in the free troposphere in winter time (Lugauer et al., 1998), and is therefore not much affected by local and near-ground emissions. The Jungfraujoch is a col between the mountains Mönch and Jungfrau, such that locally the air masses can arrive only from two directions: From north-west over the Swiss Plateau (wind direction of approx. 315 °) or from south-east over the Inner Alps via the Aletsch Glacier (approx. 135 °) (Hammer et al., 2014).

IPR were sampled by the Ice-CVI from orographic, convective and non-convective clouds. Under cloud conditions the ALABAMA was connected to the Ice-CVI, whereas during cloud-free conditions, the instrument sampled through a heated total aerosol inlet (total; 20 °C; Weingartner et al., 1999). Both inlets were installed on the roof of the Sphinx Laboratory.

Under cloud conditions the ALABAMA sampled through the Ice-CVI, whereas under cloud-free condition it was switched manually to the total aerosol inlet.

The connection to the two inlet systems limited the maximal particle size of the particles reaching the ALABAMA to approximately 3 μm. The ALABAMA sampled through ¼" stainless steel tubes with different lengths (Ice-CVI to ALABAMA: 126 cm, total to ALABAMA: 261 cm). Particle losses inside the sampling tube were calculated with a modified version of the Particle Loss Calculator (von der Weiden et al., 2009). For both sampling lines the transmission efficiency was about 99 % for particle sizes between 200 nm and 500 nm and decreased to 95 % for particle sizes up to 1000 nm. The upper 50 %-cut-off of the Ice-CVI was at about 4900 nm and for the total inlet about 3300 nm.

Due to technical problems with the mass spectrometer only the cation mass spectra are available from this field deployment.

**2.4.2 Ice particle residual sampling**

The Ice-CVI (Mertes et al., 2007) was designed to sample small, fresh ice particles (< 20 µm) out of mixed-phase clouds. Such small ice crystals have grown only by water vapor diffusion and have an age of less than 20 seconds (Fukuta and Takahashi, 1999). Therefore, it is very likely that these ice crystals have formed in the vicinity of the inlets and had only little time to scavenge interstitial aerosol particles, such that the IPR extracted from such fresh ice crystals represent to a high degree to the original IPN (Mertes et al., 2007 and references therein). A detailed description and instrumental characterization is provided in Mertes et al. (2007), therefore the system is described here only briefly: The Ice-CVI consists of three main separation sections (omnidirectional inlet, virtual impactor (VI) and pre-impactor (PI)) and a CVI (counterflow virtual impactor). The omnidirectional inlet transfers particles with a particle size up to 20 µm from the aspired air without influences of precipitation and wind. To remove larger particles which entered the inlet system owing to precipitation or wind and to get a defined upper sampling size, the VI is located just below the inlet with an upper transmission limit of 20 µm. Particles larger 20 µm are virtually impacted while smaller particles remain in the sample flow. Afterwards, ice crystals are separated from the supercooled droplets with the help of the pre-impactor (two-step separation system with 10 µm and 4 µm impaction stages). The impaction plates of the pre-impactor are not actively cooled but adopt ambient temperature which must be below 0°C to allow for mixed phase clouds to exist. The small ice particles bounce off the plates and remain in the sample flow whereas the supercooled droplets freeze on the plates upon contact. The transmission efficiencies of the pre-impactor with respect to supercooled droplets and ice crystals are close to 0 % respectively 100 % (Tenberken-Pötzsch et al., 2000; Mertes et al., 2007). Subsequently, the CVI removes all particles smaller than 5 µm, i.e. the interstitial aerosol and smaller supercooled droplets and small ice crystals fragments that are possibly still in the sampling flow. To accelerate the arriving air flow to 120 ms$^{-1}$ the CVI is located inside a wind tunnel behind the VI and PI. This velocity is required to achieve a size cut of approximately 5 µm. Only particles with sufficient inertia are able to overcome the counterflow inside the CVI. Consequently, only ice crystals with an aerodynamic diameter between 5 µm and 20 µm are sampled. The collected ice crystals are injected into a particle free and dry air inside the CVI, where the ice is completely evaporated. The released particles are the IPR and are transferred to different measurement instruments for physical and chemical characterization.

The sampling principle of the Ice-CVI leads to an enrichment of the sampled particles, which is calculated by the flow ratio before and inside the CVI inlet.

A condensation particle counter (CPC, Type 3010, TSI Inc.) is located behind the CVI and measures the INP number concentration

**3    Results and Discussion**

**3.1   Laboratory measurements of reference particles**

A summary of all investigated particles types (subdivided into three classes "biological", "mineral", and "anthropogenic") is provided in Table 2 with their characteristic marker peaks. There are certain particle types where the number of mass spectra containing characteristic and unique marker peaks is relatively low (e.g.

grounded maple leaves, brown coal, desert dust and volcano dust). This results partially in high uncertainties in the identification of these particle types in ambient data

Table 2 shows that some particle types belonging to one class show similarities in their marker peaks. For instance, the biological particle types bacteria and pollen have very similar fragmentation patterns (m/z -45 ($[C_2H_5O/CHO_2]^-$), -63 ($PO_2^-$), -71 ($[C_4H_7O/C_3H_3O_2]^-$), -79 ($PO_3^-$) and 47 ($PO^+$); fragments of oxidized organic carbon and phosphate). Also cellulose (microcrystalline) and ground leaves exhibit similar marker peaks (m/z 18 ($NH_4^+$)/($H_2O^+$), 30 ($[CH_4N]^+$/$[COH_2]^+$), 58 ($[C_3H_8N]^+$/$[C_3H_6O]^+$); fragments indicating an amine-like or oxidized organic structure). Thus, it is not possible to distinguish here between different types of biological aerosol particles. Nevertheless, in general the identification of biological aerosol with the help of characteristic marker peaks is possible.

Additionally, also similarities between particle types from two different classes occur: Sea salt (industrial produced; Sigma Aldrich) and particles from cooking/barbecue emissions have similar fragmentation patterns in the cation spectra (m/z 46, 81, 83, 97, fragments of sodium/potassium components).

Cigarette smoke produced in two different ways was also measured: smoldering cigarette smoke and cigarette smoke which was firstly inhaled. The particles from smoke after inhalation do not show the characteristic marker peaks that were observed for PAH particles in the laboratory study. But neither type of cigarette smoke could be unambiguously identified.

It was observed that some rare fragmentation patterns from pollen and biomass burning particles show similarities within the cation spectra (only one sodium (m/z 23) and potassium (m/z 39) peak). This is most likely due to the non-uniform laser ablation and ionization process, leading to production of only those two ions. Summarizing, it was found that in general the presence of both polarities is of great importance for an unambiguous identification of a specific particle type. Only few particle types, as for example the anthropogenically produced particle types show distinct marker peaks in the cation spectra that are sufficient for identification.

It has to be noted that matrix effects may complicate the identification of particle types by markers peaks. Here we have only analyzed pure substances (with exception of the source sampling types and the natural dust samples). But in laser ablation mass spectrometry, the ionization efficiency can be a function of the particle matrix (e.g., Gross et al., 2000) such that marker peaks of certain particle types might be less abundant in internal particle mixtures. Future studies will therefore also include reference mass spectra from various types of mixed particles.

### 3.2 Results on IPR composition and out-of-cloud aerosol at the Jungfraujoch
### 3.2.1 Identified Particle Types

Altogether 71064 background aerosol particles and 595 IPR were analyzed during 217 h and 111 h measurement time, respectively. For the identification of specific particle types the marker peaks that resulted from the laboratory studies were applied to the Jungfraujoch data. Although, as mentioned above, the presence of both polarities allows in general for a better classification, the application of the marker peaks only for the cations also yielded useful results, because many distinguishing characteristics are found in the cation spectra (Table 2). In this way 13 different particle types were identified. The average spectra of each particle type with the highlighted marker peaks are shown in Fig. 2.

The particle types "biomass burning" and "soot" show both the typical $C_n$-fragmentation ($C_1 - C_7$, m/z 12 … 84)

and can be distinguished by the presence of the peak at m/z 39 ($K^+$) in the cation spectra of the particles from

biomass burning.

The particle types that were assigned to the type "engine exhaust" also show $C_n$-fragmentation ($C_1^+ - C_5^+$, m/z

12 … 60) but can be distinguished by the peak at m/z 40 ($Ca^+$) which was observed in the reference mass spectra

but also previously by other researchers (Trimborn et al., 2002; Vogt et al., 2003; Shields et al., 2007).

PAH containing particles were identified through the corresponding reference spectra and marker peaks from the

laboratory studies, namely 50/51 ($C_4H_{2/3}^+$), 63 ($C_5H_3^+$), 77 ($C_6H_5^+$), and 91 ($C_7H_7^+$). Even though cigarette

particles (before inhalation) contain these markers as well, our reference spectra indicate that these two particle

types can be distinguished because cigarette smoke additionally contains a $C_n$ pattern (m/z 12 − 36).

Two different fragmentation patterns of biological particles were found during the campaign. One type shows

the marker peaks at m/z 18, 30, 58 and 59, which indicates an amine-like or oxidized organic structure. The other

one shows the marker peak at m/z 47 ($PO^+$).

Additionally, soil dust was identified based on the laboratory studies. It is characterized by the presence of

mineral components mixed with organic, biological material (e.g. peaks at m/z 18, 30, 58 and 47 point to

biological components).

The laboratory data have shown that particles produced from cooking/barbecue emissions and sea salt particles

have the same cation fragmentation pattern in the positive ion spectra. Thus, both particle types cannot be

distinguished in this data set and therefore were merged.

Also from the particle type "aged material" two different fragmentation patterns were found. The first one shows

peaks at m/z 27 and 43 ($C_2H_3^+$ and $C_3H_7^+$; fragments of organic material related to secondary organic aerosol)

with a high relative intensity. The other one shows peaks at m/z 92, 108 and 165 ($Na_2NO_2^+$, $Na_2NO_3^+$, $Na_3SO_4^+$),

indicating aged sea salt processed by nitrate and sulfate containing compounds (Gard et al., 1998).

The particle type "industrial metals" is marked by peaks of metal ions typically occurring in urban or industrial

emissions (e.g. m/z 51/67 ($V^+/VO^+$), m/z 54/56 ($Fe^+$), m/z 55 ($Mn^+$), m/z 58/60 ($Ni^+$), m/z 59 ($Co^+$) and m/z

63/65 ($Cu^+$) (de Foy et al., 2012)). Chromium and nickel containing particles might also originate from

contamination by the stainless steel tubes. But due to the low flow velocity and the laminar flow inside the tubes

the production of particles by abrasion from the tube walls through collision of the aerosol particles with the

inner wall of the tubes can be neglected. Another source of such contamination might be the valves that might

mechanically produce particles during opening and closing. However, such particles are expected to be detected

by the mass spectrometer within a few seconds after operation of a valve which was not the case. Thus we

consider these particles to be real ambient atmospheric particles.

Lead containing particles show the typical isotope pattern of lead (m/z 206, 207, 208) and are internally mixed

with metallic or organic components. Previous measurements at the JFJ have shown that lead containing

particles were found in the IPR (Cziczo et al., 2009; Ebert et al., 2011). However, the main component of this

particle type is organic or metallic origin. Thus it can be assumed that lead is only contained in small amounts in

these particles. Using data from the same experiment, Worringen et al. (2015) have shown that two types of lead

particles occurred in the IPR selected by the Ice-CVI during the INUIT-JFJ campaign: large homogeneous lead particles and small particles with lead inclusions. The authors concluded that only the homogeneous lead particles are artifacts produced by mechanical abrasion from the surface of the impaction plates of the Ice-CVI. Therefore, the lead containing particles described here are not considered as artifacts of the Ice-CVI.

Mineral dust particles ("minerals") were also found in the aerosol particles sampled during the JFJ campaign. This particle type was identified based on the marker peaks from the laboratory studies as well.

The types "K dominated" and "Na + K" are subcategories of the type "other", but are not clearly assignable to a certain particle type. As we inferred from the laboratory studies both particle types could originate from biological particles (e.g. pollen) or from biomass burning. On the other hand it is also possible that the "K dominated"-type is a fragmentation pattern of an inorganic salt (e.g. $K_2SO_4$). An unambiguous classification of these particle types from cation spectra only is not possible.

Most of these particle types do not represent pure particles like those investigated during the laboratory studies. The particles contain also other substances (as can be seen in the mass spectra) but here the most prominent marker peaks were used to identify the dominating particle type.

The type "others" includes all spectra which could not be unambiguously identified as one of the introduced particle types. This may partly be due to missing reference spectra, such that a further extension of the reference data base will allow for an identification of particles in the "other" fraction, but also due to complex mixtures of particles that cannot be identified here, especially because the anions were not available.

### 3.2.2 IPR composition compared to out-of-cloud aerosol particles

Figure 3 shows the relative abundance of the identified particle types in all aerosol particles sampled out-of-cloud in comparison to all sampled IPR during all cloud periods. The table included in Fig. 3 gives the absolute number of particles per particle type, the percentage of this particle type, and (in the last column) the "enrichment factor", i.e. the percentage of the particle type found in IPR divided by the percentage found in the out-of-cloud aerosol.

In comparison to the out-of-cloud aerosol we find a higher number fraction of particles from primary and/or natural sources in the IPR ensemble. We attribute biological and sea salt to primary natural sources, whereas soil dust and minerals emissions can directly be influenced by anthropogenic activities and can therefore not be regarded as purely natural. Biomass burning particles generated from forest fires are not primary particles, but can be related to natural sources as well. Between about 43 and 50 % of the identified particle types can be attributed to primary and/or natural sources; the uncertainty range is mainly due to the inability to separate between sea salt particles and cooking/barbecue emissions. The general trend of this finding agrees with the only result so far in the literature on single particle mass spectrometric analysis of IPR from mixed phase clouds (Kamphus et al., 2010): Using two single particle mass spectrometers, they report from one instrument (SPLAT) that 57% of all IPR were mineral particles or mixtures of minerals with sulfate, organics, and nitrate. The other instrument (ATOFMS) reported that these two particles types represent a much higher fraction (78%) of all IPR, plus additionally 8% metallic particles. However, these data sets are based on smaller numbers of particle than our study (ATOFMS 152 particles, SPLAT 355 particles), such that here variations of air mass origin and meteorological conditions can be the main reason for such differences and none of these data sets can be regarded as representative for mixed phase clouds at the Jungfraujoch in general. A recent paper by Cziczo et al.

(2013) summarized their analyses of ice crystals sampled during various field studies. Although formation of cirrus clouds occurs under different conditions than ice formation in mixed phases clouds, it is interesting to compare these results as well. These data clearly show that mineral dust is the most dominant heterogeneous ice nucleus in almost all cirrus encounters, but that under homogeneous freezing conditions the upper tropospheric background aerosol particles (sulfate/organic/nitrate) as well as biomass burning particles are detected in the cirrus IPR. In our data the IPR population additionally shows a higher fraction of lead containing particles ($7.4 \pm 3$ %), industrial metals ($3.5 \pm 1$ %) and particles from engine exhaust ($6.4 \pm 3$ %) in comparison to the composition of the out-of-cloud aerosol. This enrichment of lead-containing particles measured at the JFJ had already been found by Cziczo et al. (2009), Kamphus et al. (2010), and Ebert et al. (2011). The out-of-cloud aerosol shows a higher fraction of aged material ($30.5 \pm 5$ %), combustion particles ($12 \pm 2$ % PAH/soot and $10 \pm 1.5$ % biomass burning) and potassium-dominated particles ($11 \pm 2$ %). From the absence of potassium-dominated particles as well as the absence of biomass burning particles within the IPR ensemble, together with the occurrence of the same fragmentation pattern in the laboratory data for biomass burning particles, it can be surmised that the potassium-dominated type possibly originates also from biomass burning particles. On the other hand potassium containing salts also may be the source of these particles, which are not acting as INP (Twohy and Poellot, 2005).

It must be taken into account that the detection of potassium in laser ablation mass spectrometry is very efficient due to its low ionization efficiency (Gross et al., 2000; Silva and Prather, 2000), such that only a small amount of potassium in a particle results in a large ion signal.

It is unexpected particles from engine exhaust but not particles from biomass burning are found in the IPR ensemble, because the latter are also assumed to have good ice nucleation ability (Kamphus et al., 2010; Twohy et al., 2010; Pratt et al., 2011; Prenni et al., 2012). Additionally, lead-containing particles and particles from engine exhaust were found in INP (Kamphus et al., 2010; Corbin et al., 2012). Due to the finding that the same relative abundance of the particle type "PAH/soot" is found in both particle populations ($12 \pm 2$ % and $12 \pm 5$ %, respectively) and the finding that biomass burning particles as well as particles from engine exhaust show an organic fragmentation pattern, further research is necessary to determine which specific property of these particle types enables their ice nucleation ability. There are only a few comparable single particle measurements reported in the literature: Measurements with the ATOFMS (Aerosol Time-of-Flight Mass Spectrometer) at the JFJ (Cziczo et al., 2009; Kamphus et al., 2010), and also aircraft measurements over North America show an amount of 5 and 10 % lead-containing particles in out-of-cloud aerosol (Murphy et al., 2007). However, only a minor amount of minerals and fly ash was found at the Storm Peak Laboratory (SPL; 3200 m a.sl.) in northern Colorado (DeMott et al., 2003b). In agreement with our data, measurements from SPL also show organic material (e.g. biomass burning particles, aged material, and PAH/soot; see Fig. 3) as the major component of the out-of-cloud aerosol (DeMott et al., 2003b; Cziczo et al., 2004a).

The finding that the chemical composition of the IPR is different from that of ~~in~~ the out-of-cloud aerosol confirms the assumption that scavenging of interstitial aerosol particles does not contribute significantly to the sampled IPR, because if interstitial particle scavenging dominated, the IPR composition would look similar to that of the out-of-cloud aerosol. The presence of aged material (in low percentage) in the IPR may be explained

by aerosol scavenging, but shows the limited influence of this process on IPR composition (2.7 ± 1.3 % in IPR in

contrast to 30 ± 5 % in out-of-cloud-aerosol).

In summary, our observations showed an enhanced presence of particles from soil dust, minerals, sea

salt/cooking/barbecue emissions, engine exhaust, lead-containing particles and industrial metals in the IPR

population compared to the out-of-cloud population. Particles from aged material, from biomass burning and

potassium-dominated particles were observed to be less abundant that in the out-of-cloud aerosol.

Some particle types occurred in the same percentage in the out-of-cloud aerosol and in the IPR ensemble,

namely PAH/soot particles and biological particles. For the latter this was unexpected, because field and

laboratory studies have shown that many biological particles are efficient INP, especially at higher temperatures

(Hoose and Möhler, 2012). One possible explanation here is that biological particles are of minor importance

due to their low abundance during wintertime at the Jungfraujoch, a hypothesis that is supported by a recent

study at Jungfraujoch using light-induced fluorescence that showed that most fluorescent particles were mineral

dust and not biological particles (Crawford et al., 2016). On the other hand, enrichment of biological particles in

IPR during a Saharan dust event was observed at the Jungfraujoch in February 2014 (Kupiszewski et al., 2016)

and also in Saharan air sampled at Izaña, Tenerife, in summer 2014 (Boose et al., 2016).

### 3.2.3 Size resolved analysis

As mentioned above, the ALABAMA also allows for a size resolved chemical analysis of the sampled aerosol

particles. Additional size information can be obtained by the OPC that was operated in parallel to the

ALABAMA at the same sampling line. Figure 4 shows the size distribution of IPR and the out-of-cloud aerosol

particles analyzed by ALABAMA (a and c) and those detected by the Sky-OPC (b and d). The ALABAMA data

include only particles for which a mass spectrum was obtained. The size distribution measured with the Sky-

OPC represents all particles detected at the total inlet and the Ice-CVI, respectively.

The size distribution of the IPR analyzed by ALABAMA (Fig. 4a, right ordinates) shows in comparison to the

out-of-cloud aerosol (Fig. 4b) a wider distribution, especially to the larger particles size (d > 1000 nm). It must

be emphasized here that the ALABAMA size distribution is not corrected for sampling and detection efficiency,

the latter being optimal around 400 nm and therefore does not represent the "real" atmospheric particle size

distribution. In contrast, we consider the size distributions measured with the Sky-OPC (Figs. 4b and 4d) as

representative. They show a decrease of the particle number concentration with increasing particle diameter for

particle diameters above 250 nm (the lower size cut of the Sky-OPC). According to Fig. 4b the larger sized IPR

(d > 1 μm) are present at a higher fraction of the total particle number than the same sizes are in the out-of-cloud

aerosol (Fig. 4d). This confirms previous findings showing that larger particles are enhanced in the IPR

population (Mertes et al., 2007; Kupiszewski et al., 2016). A comparison of the absolute numbers of particles

and the calculation of an activity curve is not possible, because the out-of-cloud aerosol particles and the IPR

(inside clouds) were measured, per definition, at different times.

The size-resolved chemical composition of the IPR (Fig. 4a), normalized for each size bin, does not show a clear

relationship between size distribution and particle type, partly caused by the low counting statistics. In the lowest

size bin (100 – 200 nm) the fraction of biological and PAH/soot is highest, while mineral particles (minerals and soil dust) are enhanced in the size range between 300 nm and 800 nm. Industrial metal particle are only present in the size range from 300 nm up to approximately 1800 nm.

The size-resolved chemical composition of the out-of-cloud aerosol shows an increased number fraction of biological particles between 200 and 400 nm and larger than 1000 nm. The number of potassium-dominated particles is decreasing with particle size while the highest number of biomass burning particles is found in the size range from 300 nm up to 1000 nm. Thus, it is unlikely that the potassium-dominated particles originate mainly from biological particles or from biomass burning. In contrast, the number of "Na+K"-particles is enriched at higher sizes (> 500 nm), suggesting another source for this particle type.

**3.2.4 Case study of a selected cloud event**

The comparison of the relative abundance of all identified particles from the out-of-cloud aerosol with that of all identified IPR exhibits significant differences. However, a comparison extending over the entire data set is limited as different meteorological conditions or air mass origins are included. For a closer look at the abundance of identified particles in the out-of-cloud aerosol and the IPR, a comparison of two shorter sample periods, representing both aerosol types, was performed. To find appropriate time periods with comparable meteorological conditions, at first temperature, relative humidity, wet-bulb temperature, and wind direction were inspected. Two closely spaced sample periods were chosen, one in clouds and the other outside, with nearly the same average temperature, relative humidity and wind direction. The meteorological parameters for the two sample periods are depicted in Fig. 5 with the corresponding air mass origin back trajectories given in Fig. 6. For this purpose the HYSPLIT model was adopted (Hybrid Single Particle Langrangian Integrated Trajectory Model, National Oceanic and Atmospheric Administration; Draxler and Rolph, 2015; Rolph, 2015) with access to the meteorological data set GDAS (Global Data Assimilation; start height: 3580 m a.s.l.; calculated time: 72 h back; start time: end of the current sampling period).

The back trajectory calculations show that the air masses of both sample periods have similar, but not completely the same origin, and –besides the two excursions to higher altitudes– also a similar altitude profile of the trajectories. The air masses arrived while rising-towards the measurement platform from north-western region via France. Further, it has to be noted that both sampling periods differ in their sampling length and their time of day: The IPR sampling period lasted almost from midnight to noon, while the corresponding out-of-cloud sampling period lasted only 72 minutes in the afternoon. This may lead to different aerosol particle population due to different emission patterns of anthropogenic particles like engine exhaust, cooking etc. However, it was not possible to find two sampling periods having the same time of day and similar meteorological conditions and air mass origins. Also, the IPR sampling period could not be shortened because a sufficient number of particles needs to be sampled for a meaningful analysis.

Figure 7 shows the abundance of identified particles in ~~composition of~~ the out-of-cloud aerosol and the IPR ensemble during these two sampling times. Although the sampling conditions during both periods were very similar, the composition of these ensembles significantly differs. The IPR ensemble shows a high content of primary and/or natural material ($77 \pm 35$ %; biological particles, soil dust, minerals and sea salt/cooking/barbecue emissions). Besides that, also anthropogenic/industrial particles are enhanced in the IPR as

the last column in Fig. 7 shows: engine exhaust by a factor of 42 (± 20), lead-containing particles by 92 (± 70) and industrial metals by a factor of 4 (± 2). In comparison to that, the out-of-cloud aerosol contains a higher fraction of particles from biomass burning (22 ± 3 %) and potassium-dominated particles (22 ± 3 %). This case study shows that the observed differences between IPR and out-of-cloud aerosol particles that were observed when looking at the whole data set (Fig. 3) cannot be explained by differences in meteorological conditions and air mass origin. The finding that primary and/or natural aerosol particles such as soil dust and minerals, but also biological particles, are enhanced in the IPR population is valid for both data sets. The large number fraction of biological particles in the IPR samples of this case study exceeds that of the total IPR sample, whereas for the out-of-cloud sample it is smaller (Fig. 3). Here the variability due to different sampling times, temperatures, and air mass origins may play a role. The high number fraction of particles from biomass burning in the out-of-cloud aerosol indicates that the air masses were most likely influenced by local emissions shortly before arrival at the measurement station, but still these biomass burning particles are not found in the IPR ensemble.

Since the differences within the particle abundance of both sampling periods cannot be explained by differences in air mass origin, we assume that the difference between the out-of-cloud aerosol and the IPR is mainly caused by the ice nucleation ability of the particles at the prevailing meteorological conditions during these sample periods.

## 4    Summary

We have conducted laboratory measurements of various types of aerosol particles in order to obtain reference mass spectra for the single particle mass spectrometer ALABAMA. The results show that there are different particle classes, which can be unambiguously differentiated from each other by using characteristic marker peaks. Uncertainties of the method arise from the finding that not all particle mass spectra from one particle type display the characteristic marker peaks. This is a result of the non-uniform ionization process of laser ablation particle mass spectrometry. For some particle types (pollen, sea salt, cooking/barbecue emissions) the fraction of mass spectra showing characteristic marker peaks was high (60 – 84 %), whereas for mineral particles (desert dust, soil dust etc.) the percentage of reference mass spectra with specific markers peaks was markedly lower (20 – 37 %). The resulting particle assignment to particle types can't be corrected for this effect, but it is likely that this is the cause for the large fraction of "unknown" particles (denoted as "others") in the field results.

The derived characteristic marker peaks were applied to interpret field data where ice residuals from mixed-phase clouds were extracted by an Ice-CVI and analyzed by the mass spectrometer. The comparison of the abundance of identified particle types in the out-of-cloud aerosol particles and the IPR measured during the INUIT-JFJ campaign 2013 revealed significant differences within both ensembles. Certain particles types were found to be enriched in the IPR ensemble in comparison to the out-of-cloud aerosol. From this we can determine ambient atmospheric particle types that preferably act as ice nucleating particles under the prevailing meteorological conditions at this time. The presence of INP from lead containing particles (Cziczo et al., 2009), minerals (e.g. Hoose et al., 2010a; Kamphus et al., 2010; Hartmann et al., 2011; Hoose and Möhler, 2012; Atkinson et al., 2013), soil dust (Tobo et al., 2014), and sea salt/cooking/barbecue emissions (Wilson et al., 2015) could be confirmed. Additionally, particles from engine exhaust (Corbin et al., 2012), and industrial metals were observed to be more frequent in IPR than in out-of-cloud aerosol. It has been also reported that particles from biomass burning are efficient ice nucleating particles (Twohy et al., 2010; Pratt et al., 2011; Prenni

et al., 2012). However, during the measurements at the JFJ 2013 no particles from biomass burning were found in the IPR ensemble. In contrast to the IPR, the ensemble of the out-of-cloud aerosol particles was dominated by aged material ($31 \pm 5\%$) and particles produced by combustion ($10 \pm 1.5\%$ biomass burning and $12 \pm 2\%$ PAH/soot). The size distribution of both aerosol types have shown that the relative number of particles with a larger vacuum aerodynamic diameter measured with the ALABAMA ($d > 1000$ nm) is higher in the IPR ensemble than in the out-of-cloud aerosol. Additionally, a comparison between both particle populations was made for two closely spaced measurement periods. Although all meteorological conditions, e.g. temperature, relative humidity and wind direction (air mass origin) were similar, the chemical composition of the IPR was found to be different to that of the out-of-cloud aerosol. In comparison to the out-of-cloud aerosol particles, the IPR mainly consist of biological particles ($49 \pm 20\%$) and soil dust ($19 \pm 8\%$) whereas the ensemble of the out-of-cloud aerosol particles is enriched with particles from biomass burning ($22 \pm 3\%$) and potassium dominated particles ($22 \pm 3\%$). Because the percentage of biological particles is similar in the out of-cloud and IPR ensembles we can conclude that biological particles are ice-active at temperatures around -20 °C (temperature range between -27 °C and -6 °C over the whole measurement campaign). On the other hand, the case study indicates also a high event-to-event variability. The high number fraction of particles from biomass burning, which are not found in the IPR, indicates an influence of local emissions. This case study confirmed that the observed general differences between particle types identified in IPR and out-of-cloud aerosol is not due to different air mass origin or meteorological conditions but reflects the different ice nucleation abilities of certain atmospheric particles types. The data also show that laboratory results on the ice nucleation ability of certain particles types (e.g., mineral dust and other primary particles; Möhler et al., 2007; Hoose and Möhler, 2012; Atkinson et al., 2013; Augustin-Bauditz et al., 2014; Hiranuma et al., 2015) can at least partly be transferred to ambient atmospheric data. Some of the IPR results may be influenced by scavenging of interstitial aerosol particles by the ice crystals, but this process cannot explain the differences in the abundance of particle types between the IPR and the out-of-cloud aerosol.

**Acknowledgements**

This work was supported by the DFG projects FOR 1525 (INUIT), SPP 1294 (HALO, grant ME 3524/1-2), the Max Planck Society, the European Union Seventh Framework Programme (FP7/2007-2013) under grant agreement no 2662254 (ACTRIS TNA) and the Swiss National Science Foundation (200021L 135356).

The authors gratefully acknowledge the NOAA Air Resources Laboratory (ARL) for the provision of the HYSPLIT transport and dispersion model and/or READY website (http://www.ready.noaa.gov) used in this publication.

We would like to thank Swiss Meteorological Institute (MeteoSwiss) for providing meteorological measurements and the International Foundation High Altitude Research Station Jungfraujoch and Gornergrat (HFSJG) for the opportunity to perform experiments at the Jungfraujoch. Additional thanks go to Oliver Appel (MPIC Mainz) for help with the OPC data evaluations, to Oliver Schlenczek (University Mainz) for cloud observation at the JFJ and to Udo Kästner (TROPOS) for his help during the measurements at the JFJ.

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

1  **Table 1: Investigated particles types with the corresponding generation procedure, manufacturer and purity where**
2  **applicable data.**

| Particle type | Generation procedure | Manufacturer |
|---|---|---|
| Bacteria | AIDA (suspension) | |
| Ground leaves | AIDA (mechanically dispersed) | |
| Pollen | AIDA/washing-water | |
| Cellulose | Mechanically dispersed | Sigma Aldrich |
| Sea salt | Solution | Sigma Aldrich |
| Biomass burning | Combustion (chimney) | |
| Brown coal | Combustion (chimney) | |
| Cigarette smoke | Combustion (closed room) | |
| Cooking/barbecue emissions (sausage, steak ,cheese) | Directly sampled during a barbecue (courtyard) | |
| Engine exhaust | Directly sampled at the exhaust pipe | |
| PAH: | Suspension | |
| Benzo[ghi]perylene | | Fluka (level of purity $\geq$ 98 %) |
| Triphenylene | | Fluka (level of purity $\geq$ 98 %) |
| Dibenzo(a,h)anthracene | | SUPLECO Analytical (99.9 % purity) |
| Soot | AIDA (combustion) | |
| Mineral | AIDA (suspension) | |
| Desert dust | AIDA (suspension) | |
| Soil dust | AIDA (suspension) | |
| Volcano dust | AIDA (suspension) | |
| Additionally investigated biological particles | | |
| Alanine | Solution | Roth (purity $\geq$ 99 %) |
| Cysteine | Solution | Sigma Aldrich (purity 97 %) |
| Glutamic acid | Solution | Alfa Aesar (purity 99 %) |
| Leucine | Solution | Fluka (purity > 99 %) |
| Proline | Solution | Roth (purity $\geq$ 98.5 %) |
| Tryptophan | Solution | Roth |
| Valine | Solution | Roth (purity $\geq$ 98.5 %) |
| Glucose | Solution | Roth (purity $\geq$ 99.5 %) |
| Sucrose | Solution | Roth (purity $\geq$ 99.5 %) |
| Riboflavin | Suspension | Acros Organics (purity 98 %) |
| Chlorophyll | Suspension | Roth |
| Hemoglobin | Solution | Sigma Aldrich |

**Table 2: Overview of the different measured particle classes from primary biological, sea salt, combustion and mineral sources with their characteristic marker peaks, along with number and percentage of spectra which include these marker peaks. Peaks marked in red are the characteristic marker peaks of each particle class, peaks marked in blue show the characteristic marker peaks of one particle type. "−" designates anion spectra and "+" cation spectra.**

| Particle class | Particle type | Marker peaks [m/z] | Number of mass spectra with marker peaks | Comments |
|---|---|---|---|---|
| biological/amine | Bacteria | -: 16 [O], 26 [CN, $C_2H_2$], 42 [CNO, $C_2H_2O$, $C_3H_6$], **45 [$C_2H_5O$]**, **63 [$PO_2$]**, **71 [$C_3H_7O$, $C_3H_3O_2$]**, **79 [$PO_3$]**, 96 [$SO_4$], 97 [$HSO_4$] <br> +: **23 [Na]**, 39 [K], **47 [PO]**, 56 [Fe], **97 [NaKCl, $C_7H_{13}$]** | 1042 (42 %) | Snomax[®] shows no Peak at m/z +56 [Fe, CaO] and -97 [$HSO_4$] |
| | Ground maple leaves | -: 62 [$NO_3$], 97 [$HSO_4$], **125 [$H(NO_3)_2$]**, **195 [$H(SO_4)_2$]** <br> +: $C_n$: 12-36, **18 [$NH_4$]**, 27 [Al, $C_2H_3$], **30 [$CH_4N$]**, 39 [K], **58 [$C_3H_8N$]** | 93 (48 %) | |
| | Pollen | -: 26 [CN, $C_2H_2$], 42 [CNO, $C_2H_2O$, $C_3H_6$], **45 [$C_2H_5O$]**, **59 [$C_3H_9N/C_3H_7O$]**, 63 [$PO_2$], **71 [$C_3H_7O$, $C_3H_3O_2$]**, **79 [$PO_3$]**, 97 [$HSO_4$] <br> +: 15 [$CH_3$], 23 [Na], 39 [K], 40 [Mg], 47 [PO], 58 [**$C_3H_8N$**], **59 [$C_3H_9N$, $C_3H_7O$]** | 1277 (61 %) | Birch pollen shows additionally peaks at m/z -63 [$PO_2$], 23 [Na], and 56 [Fe, CaO] |
| | Cellulose | -: $C_n$: 24-48, **26 [CN, $C_2H_2$]**, 42 [CNO, $C_2H_2O$, $C_3H_6$], 62 [$NO_3$], <br> +: $C_n$: 12-36, 27 [Al, $C_2H_3$], **40 [Mg]**, **56 [Fe, MgO]**, **113 [$C_8H_{17}$]**, **115 [$C_9H_7$, $C_7H_{15}O$]** | 196 (18 %) | Microcrystalline cellulose shows different fragmentation pattern: m/z – 71 [$C_3H_7O$, $C_3H_3O_2$], **-125 [$H(NO_3)_2$]**, **-195 [$H(SO_4)_2$]**, **18 [$NH_4$]**, **30 [$CH_4N$]**, **58 [$C_3H_8N$]** (106 spectra of 454 (23 %)) |
| sea salt | Sea salt | -: 24 [$C_2$], 45 [$C_2H_5O$], 60 [$C_5$] **95 [$CH_3SO_3$, $PO_4$]**, 96 [$SO_4$], 97 [$HSO_4$], **99 [$H^{34}SO_4$, $NaCO_4$, $C_6H_{11}O$]**, 135 [$KSO_4$], 158 [$NO_3SO_4$] <br> +: **23 [Na]**, 24 [$C_2$, Mg], 39 [K], 40 [Ca], **46 [$Na_2$]**, **81 [$Na_2Cl$]**, **83 [$Na_2Cl$]**, 97 [$HSO_4$], **139 [$Na(NaCl)_2$]** | 173 (84 %) | |
| Combustion | Biomass burning | -: **$C_n$: 24-144**, **26 [CN, $C_2H_2$]**, 79 [$PO_3$], 97 [$HSO_4$] <br> +: **$C_n$: 12-192**, **23 [Na]**, **39 [K]** | 7436 (29 %) | |
| | Brown coal | -: **$C_n$: 24-132**, 26 [CN, $C_2H_2$], **80 [$SO_3$]**, 97 [$HSO_4$] <br> +: **$C_n$: 12-132**, **23 [Na]**, **39 [K]** | 53 (54 %) | |
| | Cigarette smoke | -: 26 [CN, $C_2H_2$], 42 [CNO, $C_3H_6$], 46 [$NO_2$] <br> +: **$C_n$: 12-36**, 27 [Al, $C_2H_3$], 39 [K], **50 [$C_4H_2$]**, **51 [$C_4H_3$]**, **63 [$C_5H_3$]**, **77 [$C_6H_5$]**, **115 [$C_9H_7$]** | 13017 (35 %) | Measurements after smoke inhalation show no PAH-fragmentation |
| | Cooking/barbecue emissions | -: 26 [CN, $C_2H_2$], 42[CNO, $C_3H_6$], 46 [$NO_2$], 97 [$HSO_4$] <br> +: 23 [Na], 39 [K], **46 [$Na_2$]**, **81 [$Na_2Cl$]**, **83 [$Na_2Cl$]**, **97 [NaKCl]**, **113 [$K_2Cl$]** | 299 (60 %) | |
| | Engine exhaust | -: **$C_n$: 24-60**, 26 [CN, $C_2H_2$], 46 [$NO_2$], 62 [$NO_3$], 79 [$PO_3$], **80 [$SO_3$]**, 97 [$HSO_4$] <br> +: **$C_n$: 12-60**, 23 [Na], 27 [Al, $C_2H_3$], 39 [K], **40 [Ca]** | 470 (40 %) | Incomplete combustions having weaker $C_n$-fragmentation and no peak at m/z -80 [$SO_3$] |
| | PAH | -: 26 [$C_2H_2$], 79 [$PO_3$], 97 [$HSO_4$] | 419 (37 %) | |

| | | | | |
|---|---|---|---|---|
| | Soot | +: 27 [$C_2H_3$], **50/51 [$C_4H_{2/3}$]**, **63 [$C_5H_3$]**, **77 [$C_6H_5$]**, **91 [$C_7H_7$]**<br>-: **$C_n$: 12-156**, 26 [CN, $C_2H_2$] | 190 (41 %) | |
| Mineral | Minerals | +: **$C_n$: 12-144**<br>-: $C_n$: 24-48<br>+: **$C_n$: 12-36**, **27 [Al]**, **40 [Ca]**, 48 [Ti], 50 [Cr], **56 [Fe, CaO]** | 827 (22 %) | |
| | Desert dust | -: $C_n$: 24-48, 26 [CN, $C_2H_2$], 42 [CNO, $C_3H_6$], **59 [$C_3H_7O$, $AlO_2$]**, **60 [$C_5$, $SiO_2$]**, **76 [$SiO_3$]**<br>+: **$C_n$: 12-36**, 7 [Li], **27 [Al]**, **40 [Ca]**, 48 [Ti], 54 [Fe], **56 [Fe, CaO]**, 64 [TiO] | 60 (20 %) | |
| | Soil dust | -: 26 [CN, $C_2H_2$], 42 [CNO, $C_3H_6$), **59 [$C_3H_7O$, $AlO_2$]**, **60 [$C_5$, $SiO_2$]**, 63 [$PO_2$], **76 [$SiO_3$]**, **79 [$PO_3$]**<br>+: 7 [Li], **27 [Al]**, 48 [Ti], 54 [Fe], **56 [Fe]**, 64 [TiO] | 721 (33 %) | Soil dust from Switzerland „Bächli" exhibits different fragmentation pattern in the anion spectra; cation spectra show additionally m/z 18 [$NH_4$], 30 [$CH_4N$], 58 [$C_3H_8N$] and only m/z 27 [Al] and 56 [Fe, CaO] of the indicated marker peaks (891 of 2122 spectra (42 %)) |
| | Volcano dust | -: 24 [$C_2$], 36 [$C_3$], 97 [$HSO_4$]<br>+: **$C_n$: 12-36**, 23 [Na], **27 [Al]**, 28 [Si], 39 [K], **40 [Ca]**, **56 [Fe, CaO]** | 32 (37 %) | |

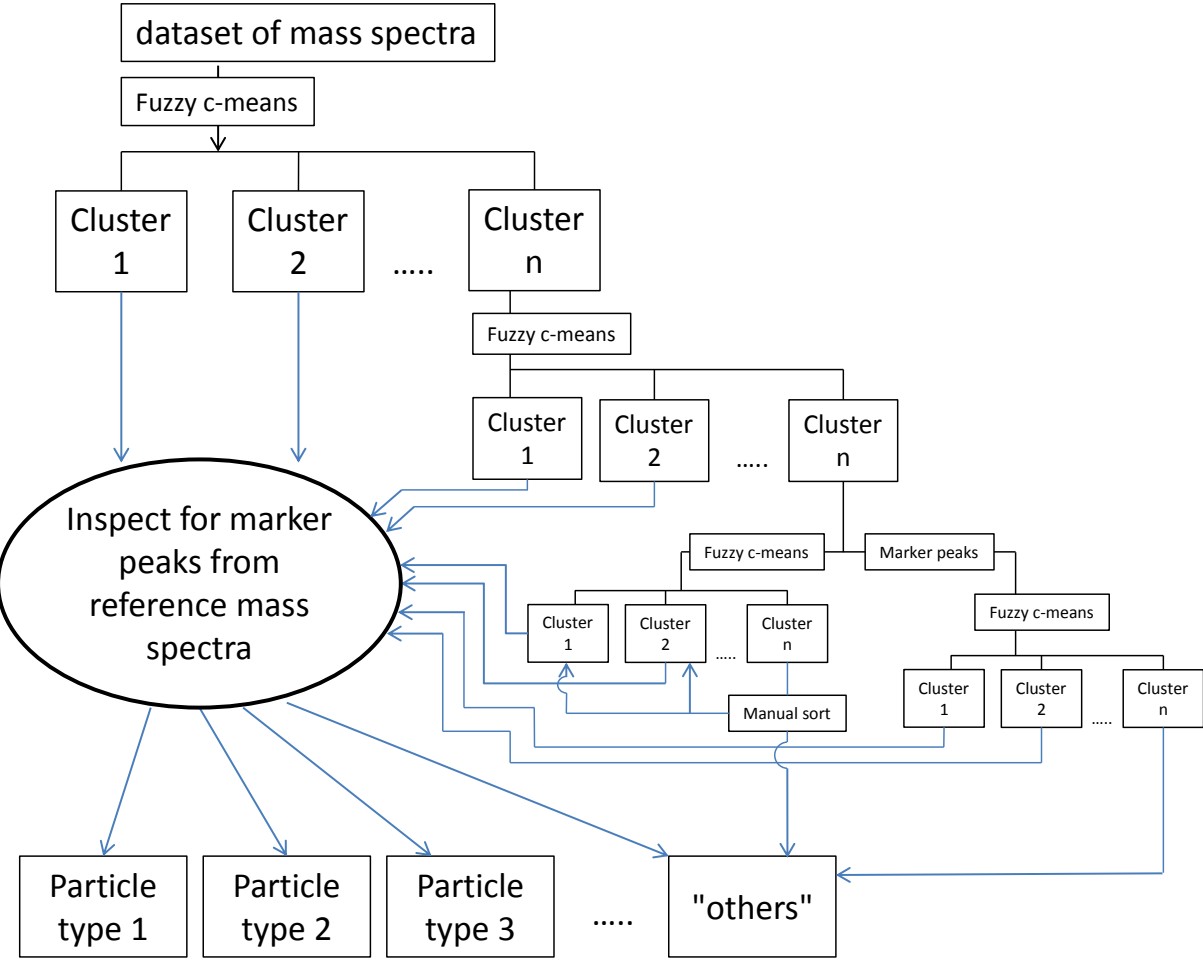

**Fig. 1: Flow chart of the single particle mass spectra evaluation. Each cluster that was found by the fuzzy c-means**

**algorithm was manually inspected for marker peaks and assigned to the appropriate particle type. This procedure**

**was repeated with the group of mass spectra (cluster n) that did not meet the distance criterion in the first clustering**

**run. Eventually the remaining mass spectra were manually sorted and inspected for marker peaks.**

**biomass burning**

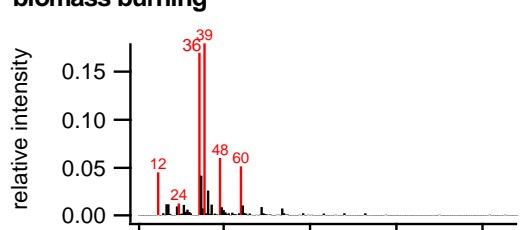

**soot**

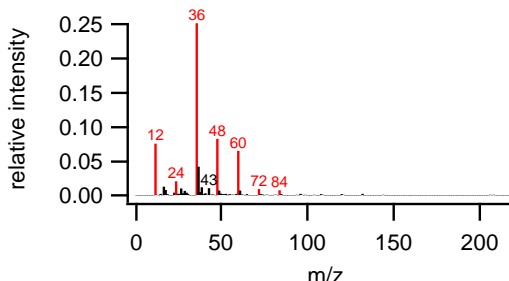

**biological particles type 1**

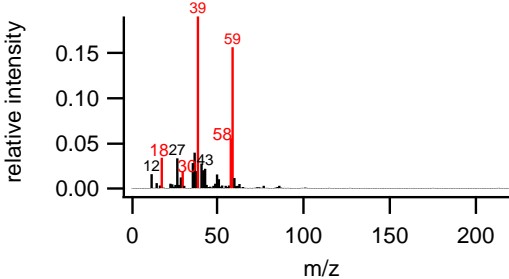

**biological particles type 2**

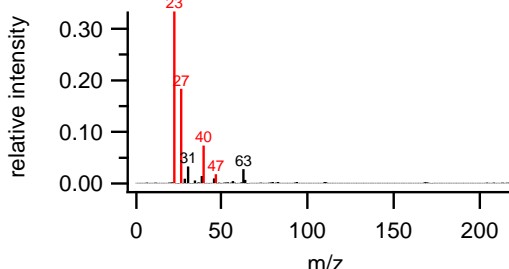

**soil dust**

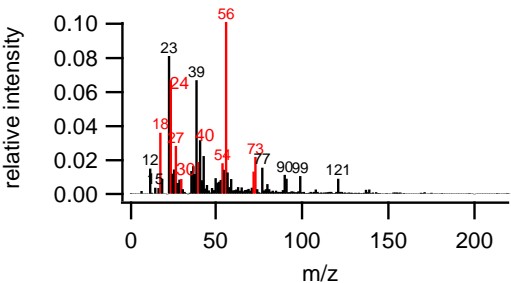

**sea salt/cooking emissions**

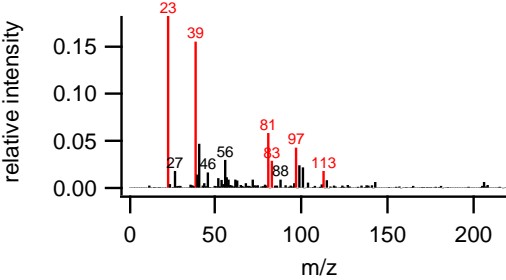

**aged material type 1**

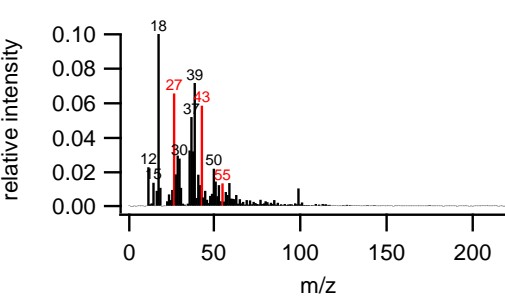

**aged material type 2**

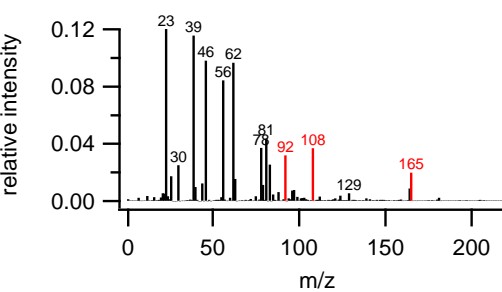

**engine exhaust**

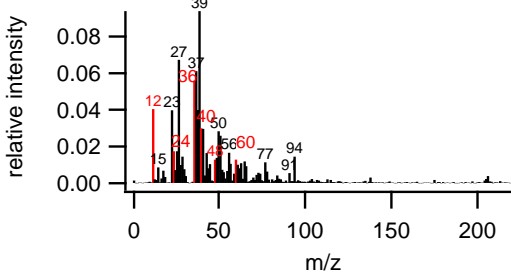

**PAH**

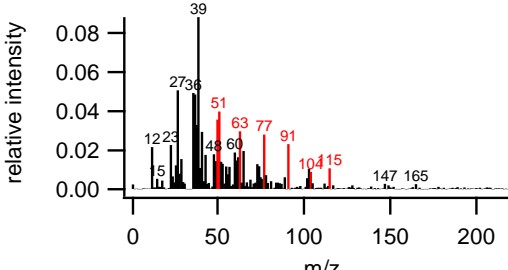

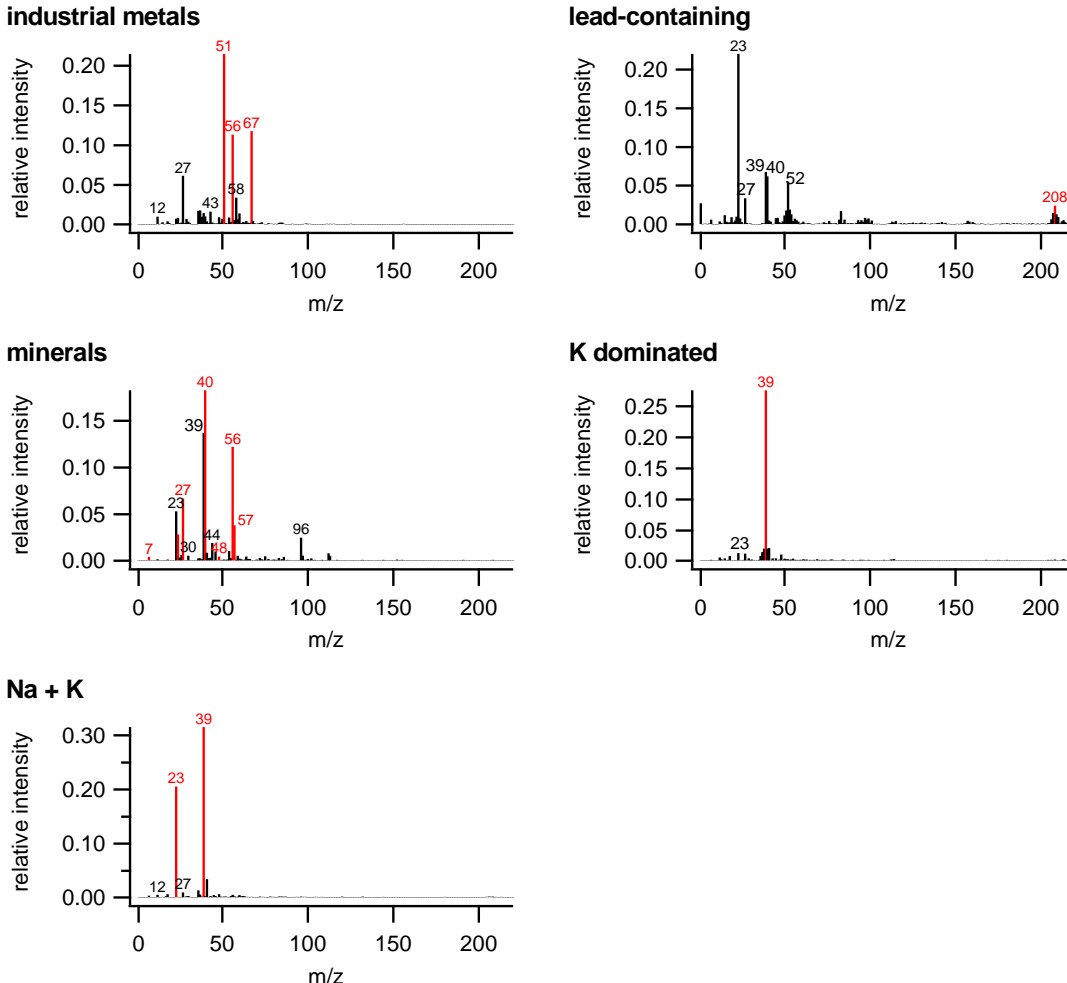

Fig. 2: Average spectra (only cations) of all identified particle types from the JFJ-measurements. The classification was done according to the results from the laboratory studies (Table 2). The red highlighted peaks indicate the marker peaks used for identification of the particle type.

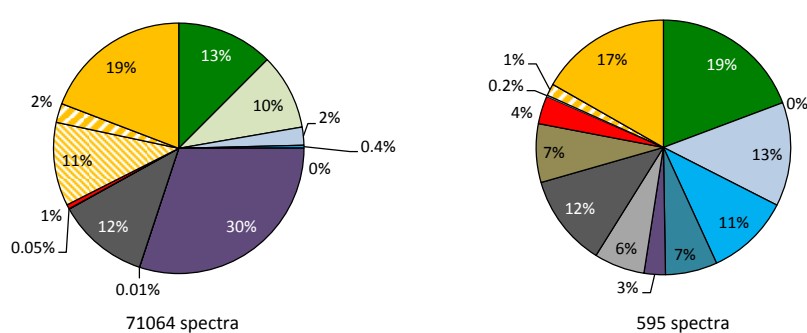

| Particle type | Out-of-cloud | | IPR | | % IPR / % Out-of-cloud |
| --- | --- | --- | --- | --- | --- |
| | Total number | percentage | Total number | percentage | |
| biological/amine | 8871 (± 1334) | 12.6 (± 1.9)% | 114 (± 47) | 19.2 (± 7.9)% | 1.5 (± 0.7) |
| biomass burning | 6959 (± 1047) | 9.9 (± 1.5)% | 0 | 0 | 0 |
| soil dust | 1655 (± 251) | 2.4 (± 0.4)% | 78 (± 32) | 13.1 (± 5.4)% | 5.5 (± 2.5) |
| minerals | 261 (± 42) | 0.4 (± 0.1)% | 67 (± 28) | 11.3 (± 4.7)% | 30 (± 14) |
| sea salt/cooking emissions | 0 | 0 | 39 (± 17) | 6.6 (± 2.8)% | -- |
| aged material | 21392 (± 3212) | 30.5 (± 4.6)% | 16 (± 7) | 2.7 (± 1.3)% | 0.09 (± 0.04) |
| engine exhaust | 8 (± 3) | 0.011 (± 0.004)% | 38 (± 16) | 6.4 (± 2.8)% | 560 (± 320) |
| PAH/soot | 8423 (± 1267) | 12.0 (± 1.8)% | 69 (± 29) | 11.6 (± 4.8)% | 1.0 (± 0.4) |
| lead-containing | 36 (± 8) | 0.05 (± 0.01)% | 44 (± 19) | 7.4 (± 3.1)% | 140 (± 70) |
| industrial metals | 399 (± 63) | 0.6 (± 0.1)% | 21 (± 5) | 3.5 (± 0.9)% | 6.2 (± 1.9) |
| K dominated | 7665 (± 1153) | 10.9 (± 1.6)% | 1 (± 1) | 0.17 (± 0.17) % | 0.02 (± 0.02) |
| Na + K | 1756 (± 266) | 2.5 (± 0.4) % | 9 (± 3) | 1.5 (± 0.5)% | 0.6 (± 0.2) |
| others | 13639 (± 1899) | 18.0 (± 2.7)% | 99 (± 41) | 16.6 (± 6.9)% | 0.9 (± 0.4) |

2    **Fig. 3: Relative abundance of identified particle types in all out-of-cloud particles (left) and all IPR (right). The table**

3    **lists absolute number of particles with uncertainties, percentages, and "enrichment factor" (percentage IPR /**

4    **percentage out-of-cloud).**

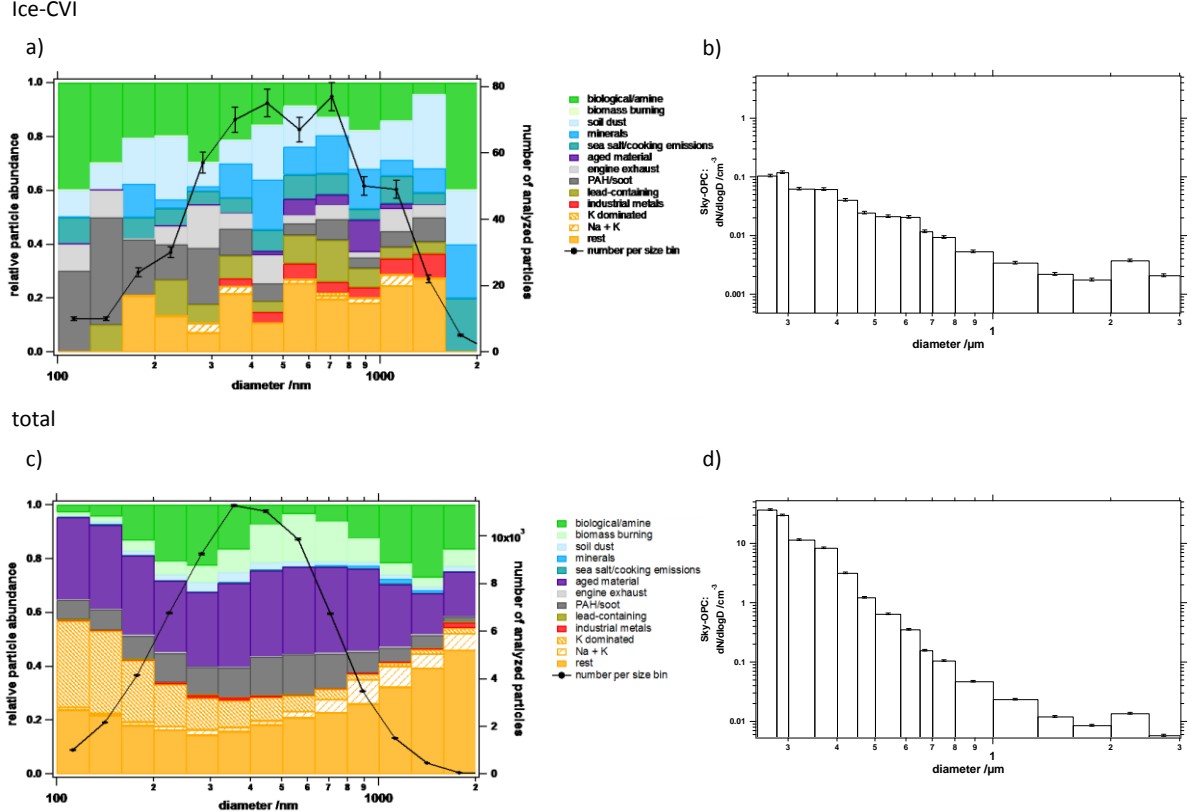

Fig. 4: Size resolved abundance of particle types in IPR (a)) and out of-cloud-aerosol particles (c)) sampled with the ALABAMA and the measured size distribution of the Sky-OPC (Ice-CVI: b; total: d). The black lines in a) and c) refer to the numbers of particles per size bin (right ordinate) of which a mass spectrum was obtained by ALABAMA with error bars based on counting statistics. The errors of the Sky-OPC data result from Gaussian propagation of uncertainty, including counting statistics, the manufacturer-given error of the OPC of 3 %, and the error of the enrichment factor (4 %).

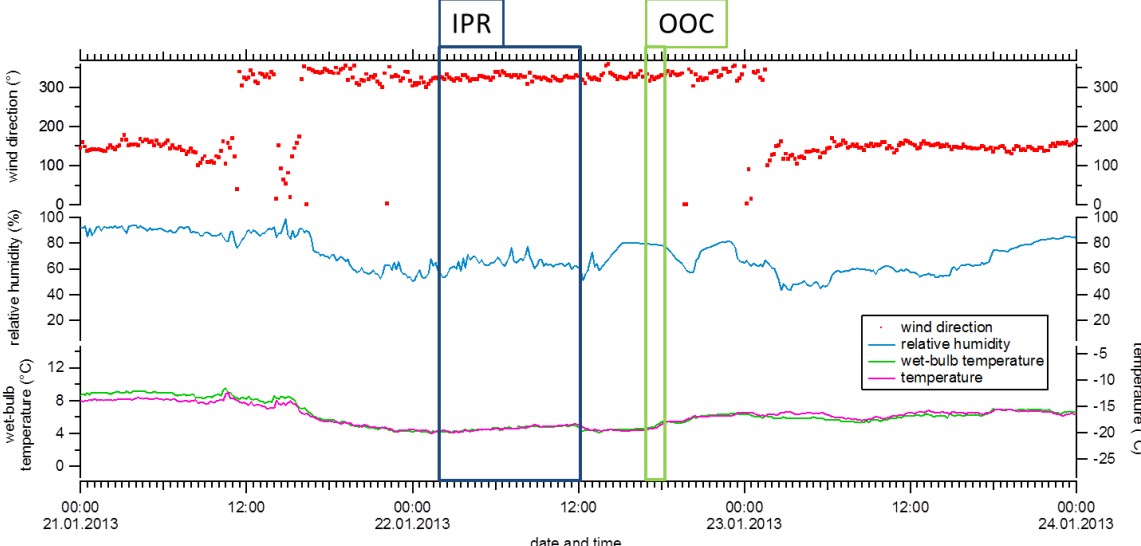

2 **Fig. 5: Wind direction, relative humidity, potential wet-bulb temperature and temperature (data from Meteo Swiss at**

3 **the JFJ). The IPR sampling period is highlighted in blue, the out-of-cloud aerosol (OOC) sampling period in green.**

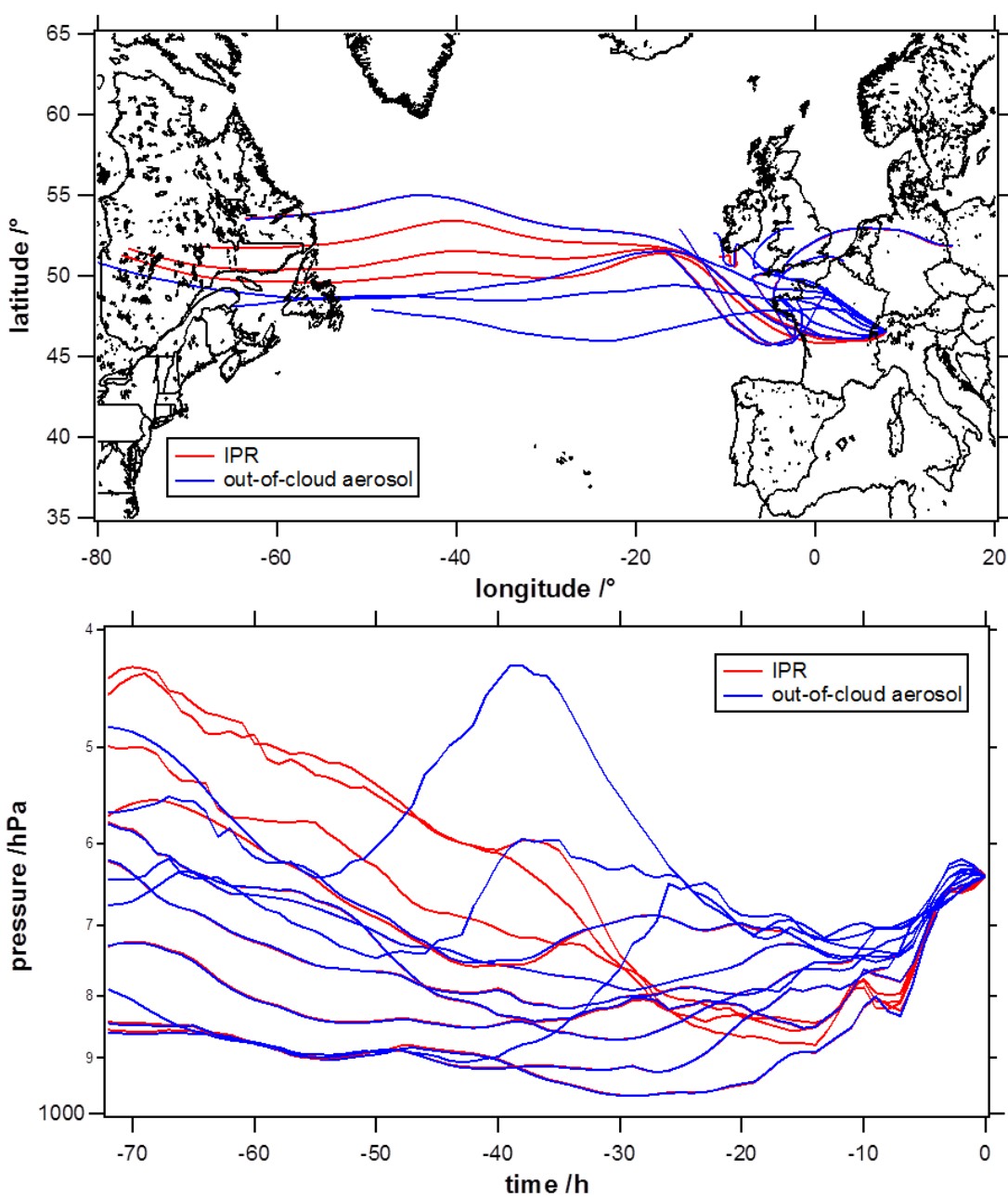

2 **Fig. 6: Back trajectories (above) and air mass pressure as a function of time (below) for both sampling periods (red:**
3 **IPR; blue: out-of-cloud aerosol).**

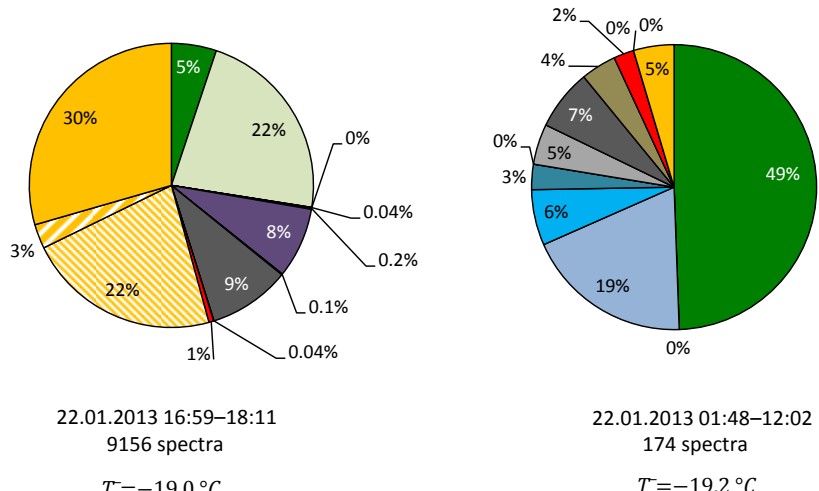

| Particle type | Out-of-cloud | | IPR | | % IPR / % Out-of-cloud |
|---|---|---|---|---|---|
| | Total number | percentage | Total number | percentage | |
| biological/amine | 469 (± 74) | 5.1 (± 0.8)% | 86 (± 36) | 49 (± 20) % | 9.6 (± 4.3) |
| biomass burning | 2047 (± 310) | 22.3 (± 3.4)% | 0 | 0 | 0 |
| soil dust | 0 | 0 | 33 (± 14) | 19.0 (±8.2) % | -- |
| minerals | 4 (± 2) | 0.04 (± 0.02)% | 11 (± 6) | 6.3 (± 3.2)% | 150 (± 100) |
| sea salt/cooking emissions | 16 (± 5) | 0.17 (± 0.05)% | 5 (± 3) | 2.9 (± 1.7) % | 16 (±11) |
| aged material | 735 (± 114) | 8.0 (± 1.2)% | 0 | 0 | 0 |
| engine exhaust | 10 (± 4) | 0.1 (± 0.04) % | 8 (± 4) | 4.6 (± 2.4) % | 42 (± 27) |
| PAH/soot | 853 (± 131) | 9.3 (± 1.4)% | 12 (± 6) | 6.9 (± 3.4) % | 0.7 (± 0.4) |
| lead-containing | 4 (± 2) | 0.04 (± 0.02)% | 7 (± 4) | 4.0 (± 2.2) % | 92 (± 70) |
| industrial metals | 50 (± 10) | 0.54 (± 0.11) % | 4 (± 2) | 2.3 (± 1.2) % | 4.2 (± 2.3) |
| K dominated | 2016 (± 306) | 22.0 (± 3.3) % | 0 | 0 | 0 |
| Na + K | 254 (± 41) | 2.8 (± 0.45) % | 0 | 0 | 0 |
| others | 2698 (± 408) | 29.4 (± 4.5) % | 8 (± 4) | 4.6 (± 2.4) % | 0.15 (± 0.08) |

Fig. 7: Comparison of the abundance of particles types in out-of-cloud aerosol (left) and the IPR (right) at similar sampling conditions during each sampling period of. The table lists absolute number of particles with uncertainties, percentages, and "enrichment factor" (percentage IPR / percentage out-of-cloud).