# Peer review of "On-line single particle analysis of ice particle residuals from mountain-top mixed-phase clouds using laboratory derived particle type assignment"

_Atmospheric Chemistry and Physics, 2016_

## Referee Comment (RC1) · Anonymous Referee #1 · 5 Jul 2016

Review of Schmidt et al., On-line single particle analysis of ice particle residuals from mountain-top mixed-phase clouds using laboratory derived particle type assignment

General comments:

The manuscript by Schmidt et al. presents valuable measurements of single-particle mass spectra from ice clouds, focusing on the composition of ice residual particles (IPR) in mixed-phase clouds. The manuscript is well formatted and presents a valuable experiment. Given the rarity of such measurements and the apparent success of the experiment, which was a difficult and complex one, the manuscript is appropriate for

publication in ACP. This manuscript is also important for having attempted a direct laboratory confirmation of each particle class.

However, I think there are several opportunities for the authors to improve the way in which the data are analyzed and reported, with respect to interpretation and statistical significance. In the following comments I will use the format p13,28 to refer to line 28 on page 13.

Major comments:

1-

The first question that comes to mind while reading this paper is the uncertainty of all reported values. Some effort was made for Fig. 3 but not for the clustering results. Roth et al. (2016, by the same group) describe a procedure to estimate uncertainties for these clustering results, why was this not applied here? If the authors have a good reason for not adapting the Roth 2016 method, and I realize some adaptation would be necessary, then they should use some other approach to numerically report their best estimated clustering uncertainties.

Clustering uncertainties should also be combined (eg in quadrature) with Poisson based sampling uncertainties to take sampling times into account. With these two sources of statistical uncertainty addressed, perhaps some of the rarer classes in Figs 2 and 6 may fall below the method limit of quantification.

Other uncertainties to be discussed include the % of laboratory particles that did not show the marker peaks and potential cross sensitivity of different marker peaks.

After uncertainties are estimated each stated percentage value % should include an uncertainty, for example those stated in Section 3.2.2.

2-

What makes this paper special compared to other single-particle mass spec papers

is the laboratory study of different particle types. Section 2.3. This must have taken a significant effort, and is well motivated. Therefore, the laboratory results should be published in full detail!

Average mass spectra for each particle type, including error bars, could be added. These could be added in the supplement and also placed next to the identified particle types in Figure 1.

The authors suggest that marker ions may be instrument specific. However, the importance or unimportance of this variability is unknown until multiple labs publish such data. Moreover, I do not see why such markers should be instrument specific if all variables (eg laser wavelength, fluence, and pulse duration) are controlled. LDI mass spectrometry databases do exist outside of aerosol science.

3-

In Section 2.3 when discussing the lab spectra it is stated "only those mass spectra that represented the majority of the different fragmentation patterns were considered". What % defines majority? Moreover, this % should be used to define a correction factor (with corresponding uncertainty propagated into the result). If only 60% of particles were measured for salt but 90% for soot, then the reported IPR numbers should be scaled up by 1/0.6 and 1/0.9 respectively.

4-

Table 1 means nothing to the reader who has not read Roth (2014), since none of the parameters were defined or explained. Roth (2014) is a PhD thesis in German, and is therefore not accessible to the general community. I have quickly looked at the other publication by Roth et al. (2016) and it looks like a great deal of effort was put into the clustering algorithm, including uncertainty consideration. So it is a pity if the reader of the present manuscript does not know that.

Could the authors please include uncertainty analysis of the clustering results in this

manuscript. Also a brief description of the conceptual basis of the chosen clustering algorithm and corresponding uncertainty are missing from Section 2.2 (p4,3).

Finally, this manuscript used the name "rest cluster" whereas Roth et al. (2016) used the name "others cluster". I find the name "rest" confusing because this word has multiple meanings. "Others" has only one..

5-

I have several comments about the marker peaks and particle types:

5a- Although the general principle of finding unique marker ions is valid, I cannot see from the manuscript how the marker ion approach was possible. On p7,2 the text states that Table 3 contains "specific" marker peaks, but since these peaks overlap in almost all cases they are not specific.

A step by step explanation of how these markers were applied is necessary to understand what was done. A flow chart would be helpful.

5b- It would be a great improvement to split the marker peaks in Table 3 into two sub-types, one containing marker peaks that are truly specific and allowed unambiguous identification of a particle type, and the other containing marker peaks that provide supporting information.

Please also clarify the meaning of the colors in Table 3. The caption says that red colors are specific to each type, but this is not true, e.g. minerals and desert dust and volcano dust all share peak 27. Bacteria and pollen share 71. etc.

5c- Cn peaks are present in cigarette smoke, fuel exhaust, soot, desert dust, volcano dust. So which peaks were used as markers to distinguish these classes? Were the minor differences in the Cn range really enough to distinguish these types? This is connected to the previous point.

5d- Please add proposed/suggested elemental formulas to each ion in Table 3. Currently only Cn peaks are identified.

5e- p8,1-3 discusses that "biological particles" showed marker peaks related to amine-like or oxidized organic structures. How have these particles been identified as biological and not simply amine-like? If the particle type cannot be unambiguously identified as biological, it should be called "biological/amine" or similar, as was done for sea salt/cooking. I don't see how these particles were recognized as biological.

5f- The ambiguity between sea salt and cooking emissions is a significant issue. Were there no complementary measurements, eg AMS or molecular markers, performed during this field study which could help to udnerstand the nature of these particles? Moreover, I don't understand why there is any ambiguity since I can see more than one unique peak for sea salt in Table 3. e.g. 135 and 158. Or could peak heights be used?

5g- how has the PAH cluster been identified? I do not see any PAHs listed in the laboratory samples and I missed a discussion in the text. It looks like some aromatic related peaks are present but are they polyaromatic?

5h- Since reference spectra of pure composition were used, the manuscript should discuss the possibility of matrix effects when internally mixed atmospheric particles were measured.

6-

p6,20 What is the probability of a small particle passing through the CVI? I can imagine that the ratio N(ice) / N(total) is similar to the ratio prob(small) / prob(large), so that small particle leakage could be a significant source of error.

7-

p11,9-14 I am not convinced by the argument that Kdominated particles were biomass burning simply because they were smaller, although it is an interesting hypothesis. Perhaps the authors could use their laboratory data to investigate this hypothesis.

8-

The larger average size of IPR is interpreted as indicative that larger particles are better INP. Has the alternative hypothesis that larger particles pass more easily through the sampling apparatus and CVI been excluded?

9-

p12,15-21. Here the authors argue that relatively more PAH/soot particles were measured during one episode because the air mass rose higher and preferentially lost other better CCN particles to wet removal. Comparing absolute instead of relative numbers would better test this hypothesis. The speculation of picking up additional local emissions after rising higher is not justified since the air mass did not fall especially low after rising higher. Either a complete and detailed analysis is needed to test this hypothesis, or the speculation should be omitted.

10-

on p12,8, "these findings agree with the general statement that natural primary aerosol such as biological particles, soil dust or minerals serve as typical ice nucleators" does not come across as a scientific. Why is this general statement being proposed? If because of recent publications, then the citations are missing.

If I am not mistaken, what the current manuscript provides is a valuable demonstration of the presence of such IPR in the field (if the statistical analysis indicates that these conclusions are robust and if proper consideration to amine or other interpretations of "biological" are given).

A better phrasing of this statement would be something like, "This case study illustrates the potentially significant contribution of biological, soil dust, minerals, and sea salt/cooking emissions on INP concentrations in mixed phase clouds, as has been identified in the laboratory (citations)".

11-

The only category for organic aerosol in this study was "aged material" which contributed 30% of the total out of cloud particle number.

It is my opinion that the chemical composition measured by ALABAMA is highly skewed relative to the chemical composition measured by quantitative techniques (e.g. AMS OA, sulfate, nitrate, and ammonium, combined with dust and EC). Presumably the ALABAMA instrument therefore has significant biases towards or against certain species. A detailed and quantitative discussion of instrument sensitivity towards different species must be included in this manuscript, if the reported pie charts are to be interpreted quantitatively.

12-

As the authors note, the present data do not allow ice nucleation rates for different particle types to be determined. Therefore please change "particles have good ice nucleating ability" to "particles were observed within ice crystal residuals" on p12,26 and please change "The high ice nucleation ability of etc etc could be confirmed" to "The presence of INP from etc etc was observed". This avoids overstating the results, which are nonetheless valuable and interesting.

The authors may also improve their manuscript by comparing the relative fractions of each particle type they have observed to previous studies of IPR or IN composition. That is, by comparing to the Cziczo citation given later in this review and the various citations already present in the manuscript. Although ice nucleation rates cannot be determined, a quantitative comparison in another dimension can nevertheless be added.

Minor:

- Percentages of particle types are frequently reported, but it is not always clear what the reference is (% of all IPR vs % relative to all of that particle type?). Please give a universal definition in Section 2. Please also add uncertainties to each reported %.

Section 3.2.1. Some particle types are given no discussion at all (engine and PAH) while others are given extensive discussion (industrial and lead). If there is some reason why engine and PAH particles were not further discussed, please make a brief note for the reader.

- p3,21 This description of LDI (laser desorption/ionization) incorrectly implies two-step vaporization and ionization. The laser does not "vaporize a fraction of created gas molecules" since ionization can occur during desorption. Better would be "the pulsed laser fires and vaporizes/ionizes the particle partly or completely."

- p3,25 what refractive index was assumed for particle sizing?

- p4,33 cooking is not combustion, so change "directly produced by combustion" to "directly sampled from the source"

- p4,33 I believe "supernatant" is the scientific term for "washing water"

p5,3 when using the word majority, please give a number

p5,29 does the manual switching mean that there is some bias in the results? Was there always a cloud free period measured at the end of the in cloud periods? Please clarify.

p6,14, what is meant by "ambient temperatures below 0 C"? Either ambient, or below 0 C.

p7,18 please define "PAH fragmentation" including citations. Can "PAH" be recognized separately from "aromatic"?

p7,20 please give the physical reason why only sodium or potassium would be observed.

p9,30 please cite a paper for this point

p9,34 I don't understand the motivation behind this sentence. The message seems to

be that a similar number fraction of biomass burning particles in the aerosol and in the IPR is an unexpected result. Why would it be unexpected?

Fig 2 and 6. Please add uncertainties for all values. Please sort the table by either out-of-cloud or IPR values. Please add a column to the table "%IPR / %out-of-cloud" to estimate the relevance enhancement of each category, including uncertainties.

-The following articles should be cited in this manuscript: Jaenicke, Abundance of Cellular Material and Proteins in the Atmosphere, Science, 308, 5718, pp 73, 2005. Cziczo et al., Clarifying the Dominant Sources and Mechanisms of Cirrus Cloud Formation, Science, 340, 6138, pp 1320-1324.

Very minor comments:

p1,20, unclear wording. try, "the outcome of these laboratory studies was particle type specific marker peaks for each investigated particle type."

p2,5 starting from "therefore" is a repetition of the previous statement which takes some thinking to realize.

p2,22 please also discuss Cziczo et al., Science 2013.

p2,28 "first results" to me is synonymous with "preliminary results". Perhaps better is "Inspection of the data set showed"

p10,35-37. "does not represent the real distribution" is more clearly phrased as "is not corrected for sampling and detection efficiency"

p11,27 with temperature and relative humidity, the wet-bulb temperature is already given?

---

## Referee Comment (RC2) · Anonymous Referee #2 · 18 Jul 2016

The manuscript by Schmidt et al. presents ambient and laboratory single particle mass spectrometer data to identify chemical components in aerosols and ice residuals on Jungfraujoch and to draw conclusions on the ice nucleation ability of different aerosol particle components. Given the importance of ice nucleation processes for precipitation and the climate, and the many uncertainties related to these processes, such studies are needed. The combination of laboratory experiments and ambient measurements makes this study especially useful for the single particle mass spectrometer community. I therefore recommend publication of this well-written manuscript in ACP after the following comments have been addressed:

[Figure]

General comments

The limitations and uncertainties of single particle mass spectrometry and the respective data should be addressed in more detail. There are several aspects:

- More care should be given as how one describes the quantities. It should be made clear, e. g. in the introduction, that "a larger relative amount" cannot be interpreted as "more aerosol mass", but can only refer to "a larger number of particles" assigned to a certain category.

- The presentation of marker mass fragments and reference spectra from different particle types is highly desirable for the single particle mass spectrometer community. For the spectra to be useful for other groups/instrument types however, information on uncertainties needs to be given. One of the issues of laser ablation single particle mass spectrometers is the (weak) repeatability of spectra/measurements; depending on particle size, chemical composition/morphology, and on the placement of the particle in the laser beam, the amount of ablated/ionized material can vary significantly. According to p. 4, l. 36, you were not size-selecting particles for your laboratory tests, which might have led to larger spectrum-to-spectrum differences. Showing e. g. standard deviations of the averaged spectra would give an idea on the uncertainty and variability of the marker spectra.

- Related to the above – how were the marker peaks identified? Given the overlap of marker peaks between different particle types, it seems to me that presenting the data as marker spectra would almost make more sense. How important are the individual markers for identification as opposed to the whole spectrum? In Table 3 you give the ratio of the number of spectra containing marker peaks (all marker peaks? just some?) to all spectra of a particular particle type. A better way to present uncertainties would be the standard deviation of the avergaed spectra (see above). The ratios in Table 3 are fairly low, I thus expect a relatively high scatter of spectra per particle type.

- More information should be given on the choice of clustering algorithm and the uncertainties of its outcome. There are a lot of references given to Roth (2014, 2016), but the main points should be conveyed to the reader in the manuscript (also in Table 1).

A general result I infer from the ambient measurements is that secondary (here in the sense of formed from a chemical reaction/in combustion) components (both organic and inorganic) or particles containing a lot of secondary (organic) components (e. g. biomass burning) are less effective ice nuclei than primary organic particles such as e. g. biological particles. In this simple categorization, however, it is hard to place the engine exhaust particles. It would be helpful if the engine exhaust and PAH particle types were discussed in the manuscript (which is not the case now).

I am confused about the similarity of sea salt and cooking emissions fragmentation pattern. How were the sea salt particles produced, what kind of cooking emissions were investigated? The variety in cooking activities is incredibly large (food cooked? what method? fuel used? etc.), and without further information a "cooking spectrum" is not very meaningful.

Soil dust and minerals are categorized as "natural" aerosol, in accordance with classifications in literature. However, soil dust aerosol concentrations can directly be influenced by anthropogenic activities (e. g. farming, mining, forestry).These are all important factors, especially when looking into/modelling anthropogenic influences on climate via aerosol-cloud interactions. I am fully aware that soil dust source apportionment lies outside the scope of this paper, but I suggest leaving out the word "natural" and potentially add a sentence on this issue.

Specific comments

P. 3 - 4, l. 37 – 2: Please give the reason for choosing the fuzzy c-means algorithm over the other two possibilities.

P. 5, l. . 3 – 9: See general comment on uncertainties and limitations of marker/spectra identification.

[Figure]

P. 5, l. 25 – 28: Was the inlet heated to 20°C to prevent condensation? Did you perform any assessment of the influence of the heating on the chemical composition (e. g. evaporation of semi-volatile material)? What was the residence time in the inlet? The evaporation of semi-volatile material could be especially important for SIA and SOA. Were parts of the ICE-CVI heated as well? Please add more information on sampling/inlet conditions.

P. 5, l.34: What about the transmission of particles with diameters larger than 500 nm? A relatively large fraction of the particles measured at Jungfraujoch was larger than 500 nm, according to Figure 3.

P. 7, l. 7 – 8: Many of the marker mass fragments of the biological particles (especially bacteria and pollen) have negative marker mass fragments. How were they identified in ambient air where you only had positive spectra?

P. 9, l. 31 – 38: See general comment above. A more thorough discussion on the properties and uncertainties of engine exhaust and PAH spectra in comparison with biomass burning spectra might shed some light on their differences in ice nucleation behavior.

P.10, l. 34 – p. 11, l. 3: Please elaborate further on the comparison and discrepancy of OPC and ALABAMA size distributions. The size distributions shown in Figure 3 are completely different and basically do not allow to draw any conclusions. Are there no artefacts of the Sky-OPC?

P.11, l. 22 – 34: Whereas the meteorological conditions were similar for the two periods, they do no coincide in termns of time of day, which however can have large influences on (anthropogenic) emission patterns (e. g. engine exhaust, cooking. . .). Please take this into account in your data interpretations.

P. 12, l. 27: This "conclusion" is a rather minor finding of your study (or put in different words, a potential reason for differences in composition between activated and nonactivated aerosol particles that can be ruled out). This sentence should not be at the end of the results section.

P. 13, l. 1 – 28: Again, in your summary, please a add a few sentences on uncertainties and the limitation of the method concerning single particle marker spectra identification and particle type detection/identification in ambient air.

Technical comments

P. 1, l. 19 – 21: "As outcome. . ." – weird sentence structure, rephrase

P. 1, l. 32: "and" instead of "an"

P. 4, l. 37: Phrase structure; should read "Coating experiments were also . . ."

P.7, l. 4 – 5: Weird sentence structure

P. 9, l. 5: Dot at the end of the sentence is missing.

P. 12, l. 15: Sentence structure: Should read "It has to be noted further..."

P. 12, l. 27: conclusion

---

## Author Comment (AC1) · 7 Oct 2016

**acp-2016-365**

**Schmidt et al., On-line single particle analysis of ice particle residuals from mountain-top mixed-phase clouds using laboratory derived particle type assignment**

Reply to Reviewer #1

Reviewer comments and questions are printed in this font type.

> Our replies are printed like this.

> Changes to the manuscript text are printed in green.

General comments:
The manuscript by Schmidt et al. presents valuable measurements of single-particle mass spectra from ice clouds, focusing on the composition of ice residual particles (IPR) in mixed-phase clouds. The manuscript is well formatted and presents a valuable experiment. Given the rarity of such measurements and the apparent success of the experiment, which was a difficult and complex one, the manuscript is appropriate for publication in ACP. This manuscript is also important for having attempted a direct laboratory confirmation of each particle class.

> Thank you for this positive rating.

Major comments:
1-
The first question that comes to mind while reading this paper is the uncertainty of all reported values. Some effort was made for Fig. 3 but not for the clustering results. Roth et al. (2016, by the same group) describe a procedure to estimate uncertainties for these clustering results, why was this not applied here? If the authors have a good reason for not adapting the Roth 2016 method, and I realize some adaptation would be necessary, then they should use some other approach to numerically report their best estimated clustering uncertainties.
Clustering uncertainties should also be combined (eg in quadrature) with Poisson based sampling uncertainties to take sampling times into account. With these two sources of statistical uncertainty addressed, perhaps some of the rarer classes in Figs 2 and 6 may fall below the method limit of quantification.
Other uncertainties to be discussed include the % of laboratory particles that did not show the marker peaks and potential cross sensitivity of different marker peaks.
After uncertainties are estimated each stated percentage value % should include an uncertainty, for example those stated in Section 3.2.2.

> We added an error estimation following the method described in Roth et al (2016).

> The method is based on a manual inspection of a subset of the data. The assignment of a certain cluster to a particle type is based on the presence of the reference marker peaks in the averaged cluster mass spectrum. Upon inspection of all mass spectra in one cluster it may occur that the marker peaks (or not all of the marker peaks) are not present in an individual mass spectrum. Such a mass spectrum has nevertheless been correctly (from a mathematical point of view) assigned to the cluster by the algorithm, because the overall correlation of the

mass spectrum with the cluster average is sufficiently high (r > 0.7). This can especially occur in cases when many other peaks are similar, as is often observed for organic particles.

For the error estimation, such particle mass spectra were regarded as "uncertain assignments". The percentage of such uncertainly assigned mass spectra was regarded as the relative error and was generalized to the whole data set.

Of the out-of-cloud data set we inspected two clusters, one assigned to biological particles (338 particles) and one that was assigned to biomass burning aerosol (473 particles). It turned out that 52 of the 338 "biological" mass spectra were uncertain (15%), and 48 of the 473 "biomass burning" mass spectra (10%). Thus, we conservatively estimated the relative error for the out-of-cloud aerosol to be about 15%.

Of the IPR data set, where the absolute numbers of particles are much lower, it was possible to do a more detailed inspection of the clusters: We inspected one cluster assigned to biological particles, where we found that 28 out of 76 were uncertain (37%), and one cluster of the "PAH/soot" particle type, where 9 out of 23 spectra were uncertain (40%). Those particle types containing only a small number of particles ("industrial metals", "Na + K", "aged material") were completely inspected manually, yielding uncertainties for the "industrial metals" of 14%, of the "Na + K" type of 0% (no uncertain particles), and of the "aged material" type of 44%.

Thus, we estimated the relative error (from uncertain particle type assignment) of the IPR population to be 40% with the exception of the industrial metals (14%) and the "Na + K" type.

These error estimates are conservative upper limits for the error range, because the reference laboratory measurements have shown that, e.g., not all biological particles contain the characteristic marker peaks. It may therefore well be that mass spectra that are similar to the cluster average spectrum of a "biological particle" type are really biological particles, even if they do not contain the marker peaks.

For the total error, this relative error was combined with the Poisson statistic error (by error propagation) of each particle type. The error ranges were added to Figure 2 (previously Fig. 1) and Figure 7 (previously Fig. 6) and in the text in all cases where the relative abundance of particles is mentioned.

The description of the error estimation given above was added to the text of the manuscript (section 2.2):

The uncertainties reported along with these numbers were estimated by manual inspection of a subset of the data, as described in Roth et al. (2016). The assignment of a certain cluster to a particle type is based on the presence of the reference marker peaks in the averaged cluster mass spectrum. Upon inspection of all mass spectra in one cluster it may occur that the marker peaks (or not all of the marker peaks) are not present in an individual mass spectrum. Such a mass spectrum has nevertheless been correctly (from a mathematical point of view) assigned to the cluster by the algorithm, because the overall correlation of the mass spectrum with the cluster average is sufficiently high (r > 0.7). This can especially occur in cases when many other peaks are similar, as is often observed for organic particles.

For the error estimation, such particle mass spectra were regarded as "uncertain assignments". The percentage of such uncertainly assigned mass spectra was regarded as the relative error. Of the out-of-cloud data set we inspected two clusters, one assigned to biological particles (338 particles) and one assigned to biomass burning aerosol (473 particles). It turned out that 52 of the 338 inspected "biological" mass spectra (15%), and 48 of the 473 inspected "biomass

burning" mass spectra (10%) had to be considered as uncertain. Thus, we conservatively estimated the relative error to be about 15% and generalized this error for whole out-of-cloud data set.

For the IPR data set, where the absolute numbers of particles are much lower, it was possible to do a more detailed inspection of the clusters: We inspected one cluster assigned to biological particles, where we found that 28 out of 76 were uncertain (37%), and one cluster of the "PAH/soot" particle type, where 9 out of 23 spectra were uncertain (40%). Those particle types containing only a small number of particles ("industrial metals", "Na + K", "aged material") were completely inspected manually, yielding uncertainties for the "industrial metals" of 14%, of the "Na + K" type of 0% (no uncertain particles), and of the "aged material" type of 44%. Thus, we estimated the relative error (from uncertain particle type assignment) of the IPR population to be 40% with the exception of the industrial metals (14%) and the "Na + K" type.

These error estimates are conservative upper limits for the error range, because the reference laboratory measurements have shown that, e.g., not all biological particles contain the characteristic marker peaks. It may therefore well be that mass spectra that are similar to the cluster average spectrum of a "biological particle" type are really biological particles, even if they do not contain the marker peaks. The uncertainty inferred from manual inspection was combined with the Poisson counting statistics error (by error propagation) for each particle type.

2-
What makes this paper special compared to other single-particle mass spec papers is the laboratory study of different particle types. Section 2.3. This must have taken a significant effort, and is well motivated. Therefore, the laboratory results should be published in full detail!
Average mass spectra for each particle type, including error bars, could be added.
These could be added in the supplement and also placed next to the identified particle types in Figure 1.

We decided to add the mass spectra of the reference particle to the supplement. Placing them in Figure 1 is not ideal, because there is usually more than one cluster for each reference particle type and we would like to show these different clusters.

Changes to text:

The laboratory reference mass spectra are shown in the supplement (Figures S1 through S20). It was found that one reference particle type produced several types of spectra which were separated using the clustering algorithm described above. The supplement lists the main clusters with the number of spectra in each cluster.

The authors suggest that marker ions may be instrument specific. However, the importance or unimportance of this variability is unknown until multiple labs publish such data. Moreover, I do not see why such markers should be instrument specific if all variables (eg laser wavelength, fluence, and pulse duration) are controlled. LDI mass spectrometry databases do exist outside of aerosol science.

We agree, if all parameters were controlled and identical, marker ions should not be instrument specific. However, custom built instruments (like our ALABAMA) have different geometries and optical components than others, such that even when the same laser type is

used, the energy density at the point of particle ablation may be different. Also the broadening of the particle beam (different aerodynamic lenses, difference flight path lengths) and thereby the variation of the ablation point, and also the acceptance region of the mass spectrometer itself (extraction voltages may be different) are not the same in different instruments. Thus, to our opinion there are many reasons why these marker ions are instrument specific, and care should be taken when using our reference mass spectra for other instruments.

3-

In Section 2.3 when discussing the lab spectra it is stated "only those mass spectra that represented the majority of the different fragmentation patterns were considered". What % defines majority? Moreover, this % should be used to define a correction factor (with corresponding uncertainty propagated into the result). If only 60% of particles were measured for salt but 90% for soot, then the reported IPR numbers should be scaled up by 1/0.6 and 1/0.9 respectively.

In the supplement we show the mass spectra types (clusters) from each reference particle type along with the percentages.

Correction of the reported IPR numbers can't be done in this way. First, we might end up with more particles that were actually measured. Second, it is only possible to assign marker ions of atmospheric particles to those reference spectra containing these marker ions. Speculating about spectra not containing the markers would only increase the uncertainty.

4-

Table 1 means nothing to the reader who has not read Roth (2014), since none of the parameters were defined or explained. Roth (2014) is a PhD thesis in German, and is therefore not accessible to the general community. I have quickly looked at the other publication by Roth et al. (2016) and it looks like a great deal of effort was put into the clustering algorithm, including uncertainty consideration. So it is a pity if the reader of the present manuscript does not know that.
Could the authors please include uncertainty analysis of the clustering results in this manuscript. Also a brief description of the conceptual basis of the chosen clustering algorithm and corresponding uncertainty are missing from Section 2.2 (p4,3).

We removed Table 1 and included the information into the text, describing the meaning of each parameter. Also, we added a general explanation of the clustering and of the chosen algorithm to section 2.2.:

Basically, a clustering algorithm tries to find the optimum number of clusters (i.e. groups of mass spectra) that represent the particle population by their average mass spectrum. By nature of the aerosol particle diversity and the non-uniform ionization in laser ablation ionization, it can't be expected that all particles contained in a cluster equal the average cluster spectrum (Hinz et al., 1999). Rather, each spectrum is assigned to that cluster where the distance metric (in our case one minus Pearson's correlation coefficient r) of the single particle spectrum and the averaged cluster spectrum reaches a minimum. The fuzzy c-means algorithm differs from the k-means in that way that it accounts for the possibility that one particle may belong to two (or more) clusters by using membership coefficients, whereas the k-means assigns each spectrum strictly to that cluster where correlation with the averaged spectrum is highest.

Here we applied the fuzzy c-means algorithm because sensitivity tests conducted in the framework of a PhD thesis (Roth, 2014) with laboratory-generated particle of known composition and number have shown that the fuzzy c-means better separates the particle types and suffers less from false assignments. Also, various parameters that influence the clustering result where tested by Roth (2014), resulting in a "best choice" that was applied here as well: All mass spectra were normalized to reduce the influence of total signal intensity, and all m/z peaks were taken to the power of 0.5 to reduce the influence of the non-uniform laser ablation ionization, thereby increasing the influence of smaller peaks and decreasing that of larger signals. The "fuzzifier", a weighting exponent used for the calculation of the membership coefficients (Bezdek, 1982; Roth et al., 2016) was set to 1.2. A high number of start clusters was chosen to assure that also rare spectra types are considered in the data evaluation (for instance for the out-of-cloud data set from the JFJ campaign 2013 a number of 200 cluster was chosen; for a known particle composition as sampled during the laboratory studies a number of 10 to 50 cluster was chosen depending on the number of spectra). The start clusters were chosen randomly from the total particle population, under the condition that the correlation coefficient (r) between two randomly picked start spectra is less than 0.7. The procedure leading from the clustering algorithm to a certain number of particle types, as illustrated in Figure 1, was as follows: After mass calibration, the spectra were clustered using the fuzzy c-means algorithm, yielding a certain number of clusters. The resulting cluster number can be lower than the chosen number of start clusters. If this is the case, the number of start clusters was sufficiently high not to suppress rare spectra types. Each cluster includes a certain number ($\geq 1$) of mass spectra based on the calculated membership and distance. From all mass spectra in a cluster an average spectrum is calculated which is used for the identification of the particle type represented by each cluster. All mass spectra which did not fulfill the distance criterion ($1 - r \leq 0.3$) compared to any of the clusters were sorted in the cluster "others". The averaged spectra of each cluster were manually examined with respect to the presence of the marker peaks derived from the reference mass spectra (Section 3.1) and assigned to a certain particle type. The "others" cluster was processed again using the fuzzy c-means algorithm, but with reduced constraints and again the resulting clusters were manually examined and, if possible, assigned to particle types. At the end all clusters of the same particle type were merged, whereas clusters that could not be assigned to a certain particle type were added to the cluster "others".

We added the uncertainty discussion (as already mentioned above) to section 2.2.

Finally, this manuscript used the name "rest cluster" whereas Roth et al. (2016) used the name "others cluster". I find the name "rest" confusing because this word has multiple meanings. "Others" has only one..

We changed the name from "rest" to "others".

5-
I have several comments about the marker peaks and particle types:

5a- Although the general principle of finding unique marker ions is valid, I cannot see from the manuscript how the marker ion approach was possible. On p7,2 the text states that Table 3 contains "specific" marker peaks, but since these peaks overlap in almost all cases they

are not specific. A step by step explanation of how these markers were applied is necessary to understand what was done. A flow chart would be helpful.

> The expression "specific" was misleading. We should better have used "characteristic" and we changed it in the manuscript. The combination of these peaks was found to be characteristic for the particle class (first column in Table 3). See also reply to comment 5b.

> We also added a flow chart (Figure 1), explaining the procedure in more detail.

5b- It would be a great improvement to split the marker peaks in Table 3 into two subtypes, one containing marker peaks that are truly specific and allowed unambiguous identification of a particle type, and the other containing marker peaks that provide supporting information. Please also clarify the meaning of the colors in Table 3. The caption says that red colors are specific to each type, but this is not true, e.g. minerals and desert dust and volcano dust all share peak 27. Bacteria and pollen share 71. etc.

> Marker peaks that unambiguously identify a certain particle type are very rare. Typically we find a combination of peaks that is characteristic to a certain particle type or particle class. As stated above, we replaced "specific" by "characteristic". The combination of the colored peaks in Table 3 is used to assign a mass spectrum to a certain particle type or particle class.

5c- $C_n$ peaks are present in cigarette smoke, fuel exhaust, soot, desert dust, volcano dust. So which peaks were used as markers to distinguish these classes? Were the minor differences in the $C_n$ range really enough to distinguish these types? This is connected to the previous point.

> This is a good example to explain the procedure. First, it has to be emphasized again that the identification of particle types for the Jungfraujoch can only rely on the cations.

> Desert dust and soil dust contain $Fe^+$ (m/z 56), the combustion related particles don't. By this, we can separate the mineral particles from the combustion particles.

> Cigarette smoke, fuel exhaust, and soot are indeed very similar. But soot particles have a large $C_n$ chain (as shown in Table 3: up to $C_{10}$). This was not observed in cigarette smoke and fuel exhaust. Both particle types showed much lower $C_n$ chains. Soot and biomass burning can be separated by the presence of $K^+$ (m/z 39). Engine exhaust particle showed a peak at m/z 40 (most likely calcium, as also reported by Shields et al. (2007) and Trimborn et al. (2002). Cigarette smoke particles show the markers similar to PAH particles (m/z 50, 51, 63, 77), but additionally the $C_n$ chain which is not contained in the PAH reference spectra.

> A particle type "cigarette smoke" was not found at the Jungfraujoch, but the types PAH and soot. Both were merged to one particle type in Figure 3.

> Changes to text:

> The particle types that were assigned to the type "engine exhaust" also show $C_n$-fragmentation ($C_1^+ - C_5^+$, m/z 12 … 60) but can be distinguished by the peak at m/z 40 ($Ca^+$) which was observed in the reference mass spectra but also previously by other researchers (Trimborn et al., 2002; Vogt et al., 2003; Shields et al., 2007).

> PAH containing particle were identified through the corresponding reference spectra and marker peaks from the laboratory studies, namely 50/51 ($C_4H_{2/3}^+$), 63 ($C_5H_3^+$), 77 ($C_6H_5^+$), and 91 ($C_7H_7^+$). Even though cigarette particles (before inhalation) contain these markers as well, our reference spectra indicate that these two particle types can be distinguished because cigarette smoke additionally contains a $C_n$ pattern (m/z 12 – 36).

5d- Please add proposed/suggested elemental formulas to each ion in Table 3. Currently only Cn peaks are identified.

We added the elemental formulas to the table (now Table 2).

5e- p8,1-3 discusses that "biological particles" showed marker peaks related to aminelike or oxidized organic structures. How have these particles been identified as biological and not simply amine-like? If the particle type cannot be unambiguously identified as biological, it should be called "biological/amine" or similar, as was done for sea salt/cooking. I don't see how these particles were recognized as biological.

We found two particle types that we assigned to biological particles (Figure 2):

Type 1 shows peaks at m/z 18, 30, 39, 58, and 59.

Type 2 shows peaks at m/z 23, 27, 40, 47.

The marker peaks found from the reference spectra (Table 2) show that ground maple leaves show exact the marker peaks 18, 30, 39, 58, while pollen show 39, 58, 59. This is the reason to assign "type 1" to biological particles. We used "amine-like" to explain the occurrence of m/z 59, which may be $(CH_3)_3N^+$, a marker for trimethylamine (e.g., Healy et al., 2014). Also m/z 18, 30, and 58 can be explained by nitrogen-containing ions ($NH_4^+$, $CH_4N^+$, and $C_3H_8N^+$). The identification of "type 2" as biological relies mainly on m/z 47 ($PO_3^+$), observed in pollen and in bacteria.

We changed the name of this particle type in Figures 3, 4 and 7 to "biological/amine".

5f- The ambiguity between sea salt and cooking emissions is a significant issue. Were there no complementary measurements, eg AMS or molecular markers, performed during this field study which could help to udnerstand the nature of these particles? Moreover, I don't understand why there is any ambiguity since I can see more than one unique peak for sea salt in Table 3. e.g. 135 and 158. Or could peak heights be used?

It is explained in the manuscript (page 4 lines 1-2) that only cations were available from the Jungfraujoch field data. Thus, the anion marker peaks at 135 and 158 could not be used. The cation markers (23, 46, 81, 83, 139) occur both in sea salt and in cooking emissions (most likely from salt contained in the spicing of the meat), so from cations alone these two particle types can't be distinguished, as was explained on page 8 in lines 7-9.

There are reasons to believe that these particle are more likely from cooking that from sea salt, because Fröhlich et al. (2015) report from Jungfraujoch the observation of local POA that resembles organic aerosol from cooking (COA), but our data base (we only observed 39 particles of this type in the IPR population and none in the out-of-cloud aerosol) is too small to draw further conclusions.

5g- how has the PAH cluster been identified? I do not see any PAHs listed in the laboratory samples and I missed a discussion in the text. It looks like some aromatic related peaks are present but are they polyaromatic?

Both Table 1 and Table 2 of our manuscript list PAH. We used benzo[ghi]perylene, dibenzo(a,h)anthracene and triphenylene. We will specify this in Table 1. Table 2 and the

mass spectra now added to the supplement show that the marker peaks like 50, 51, 63, 77, 91 are well suited to identify this particle type. Cigarette smoke (before inhalation) shows these markers as well, but that's not surprising because cigarette smoke contains PAHs. However, we did not study other aromatics up to now.

5h- Since reference spectra of pure composition were used, the manuscript should discuss the possibility of matrix effects when internally mixed atmospheric particles were measured.

We added a discussion on this important point (section 3.1). It is also desirable to investigate these matrix effects with mixture of the reference particle types presented here, but this has to be a subject of future work.

It has to be noted that matrix effects may complicate the identification of particle types by markers peaks. Here we have only analyzed pure substances (with exception of the source sampling types and the natural dust samples). But in laser ablation mass spectrometry, the ionization efficiency can be a function of the particle matrix (e.g., Gross et al., 2000) such that marker peaks of certain particle types might be less abundant in internal particle mixtures. Future studies will therefore also include reference mass spectra from various types of mixed particles.

6-
p6,20 What is the probability of a small particle passing through the CVI? I can imagine that the ratio N(ice) / N(total) is similar to the ratio prob(small) / prob(large), so that small particle leakage could be a significant source of error.

Although we do not really agree that the ratio N(ice) / N(total) is similar to the ratio prob(small) / prob(large), we are aware that there might be a chance that particles below the CVI cut-size might overcome the counterflow. This was shown for the rather new concept of a pumped CVI (PCVI) by Pekour and Cziczo (2011). According to their work, the transmission of small particles should be caused by collisions, coagulation or riding the wake of particles with diameters larger than the CVI lower cut-off size, but could also be due to the different design, which creates flow imperfections in contrast to the "classical" CVI. Up to now this issue of small particle leakage was never investigated but also never observed for the CVI design used in this study. Nevertheless, we applied the simulation results for the PCVI to estimate the contribution of small particle leakage. Pekour and Cziczo (2011) provide values for the ratio $n_{out}/(n_{in} \times N_{out})$ for different sizes and as a function of the PCVI counterflow, where $n_{out}$, $n_{in}$, and $N_{out}$ denote the concentration of small transmitted (smaller than the CVI cut-off size), small initial, and large transmitted (larger than the CVI cut-off size) particles. Their results show that for the used counterflows this ratio is on the order of $10^{-5}$ cm$^{-3}$. So, the ratio of leaked small particles ($n_{out}$) to correctly sampled large particles ($N_{out}$) is

$$n_{out}/N_{out} = 10^{-5} \text{ cm}^{-3} \times n_{in}.$$

This means that only for high background aerosol number concentrations ($10^3 - 10^4$ cm$^{-3}$), the fraction of small leaked particles could be $1 - 10$ % of the correctly sampled residual particles, but most of the time it should be below 1 % at the conditions at the Jungfraujoch where particle concentrations in winter and spring are typically below 1000 cm$^{-3}$ (Herrmann et al., 2015).

7-
p11,9-14 I am not convinced by the argument that Kdominated particles were biomass burning simply because they were smaller, although it is an interesting hypothesis. Perhaps the authors could use their laboratory data to investigate this hypothesis.

We agree, this conclusion is not valid. The size distribution of the K-dominated (Fig. 4) particles looks different than that of biological and biomass burning particles. Thus, it is unlikely that the source of these K-dominated particles is biological or biomass burning. We modified the text accordingly:

The out-of-cloud aerosol shows an increased number fraction of biological particles between 200 and 400 nm and larger than 1000 nm. The number of potassium-dominated particles is decreasing with particle size while the highest number of biomass burning particles is found in the size range from 300 nm up to 1000 nm. Thus, it is unlikely that the potassium-dominated particles originate mainly from biological particles or from biomass burning.

8-
The larger average size of IPR is interpreted as indicative that larger particles are better INP. Has the alternative hypothesis that larger particles pass more easily through the sampling apparatus and CVI been excluded?

The transmission of the ALABAMA and the OPC themselves would cancel out, but the sampling line transmissions might play a role, because the sampling line from the CVI was shorter (126 cm) than that from the total inlet (261 cm). We calculated the transmission (as already stated in the manuscript), and divided the transmission curves. The ratio transmission$_{CVI}$ to transmission$_{total}$ does not deviate significantly from unity between 100 nm (ratio = 1.005) and 1500 nm (ratio = 1.04). Thus, we can rule out that sampling losses are an issue here.

9-
p12,15-21. Here the authors argue that relatively more PAH/soot particles were measured during one episode because the air mass rose higher and preferentially lost other better CCN particles to wet removal. Comparing absolute instead of relative numbers would better test this hypothesis. The speculation of picking up additional local emissions after rising higher is not justified since the air mass did not fall especially low after rising higher. Either a complete and detailed analysis is needed to test this hypothesis, or the speculation should be omitted.

We agree with the reviewer that this assumption is too speculative and we skip this suggestion, because a detailed analysis of the air mass history would be beyond the scope of this paper.

10-
on p12,8, "these findings agree with the general statement that natural primary aerosol such as biological particles, soil dust or minerals serve as typical ice nucleators" does not come across as a scientific. Why is this general statement being proposed? If because of recent publications, then the citations are missing. If I am not mistaken, what the current manuscript provides is a valuable demonstration of the presence of such IPR in the field (if the statistical analysis indicates that these conclusions are robust and if proper consideration to amine or other interpretations of "biological" are given). A better phrasing of this statement would be something like, "This case study illustrates the potentially significant contribution of biological, soil dust, minerals, and sea salt/cooking emissions on INP concentrations in mixed phase clouds, as has been identified in the laboratory (citations)".

This is a misunderstanding. This section discusses the case study where we compared IPR and out-of-cloud aerosol for two time periods with very similar air mass origin and meteorological conditions. This sentence was meant to compare the case study (Fig 7) with the whole data set (Fig 3). We have rephrased the sentence:

This case study shows that the observed differences between IPR and out-of-cloud aerosol particles that were observed when looking at the whole data set (Fig 3) cannot be explained by differences in meteorological conditions and air mass origin. The finding that natural primary aerosol particles such as biological particles, soil dust and minerals are enhanced in the IPR population is valid for both data sets.

11-
The only category for organic aerosol in this study was "aged material" which contributed 30% of the total out of cloud particle number.
It is my opinion that the chemical composition measured by ALABAMA is highly skewed relative to the chemical composition measured by quantitative techniques (e.g. AMS OA, sulfate, nitrate, and ammonium, combined with dust and EC). Presumably the ALABAMA instrument therefore has significant biases towards or against certain species. A detailed and quantitative discussion of instrument sensitivity towards different species must be included in this manuscript, if the reported pie charts are to be interpreted quantitatively.

It has to be understood that particle type and mass concentration are two completely different ways of looking at aerosol particle population. For example, consider a uniform monodisperse particle population, where each particle contains 30% $NH_4NO_3$, 30% $(NH_4)_2SO_4$, and 40% organic matter, the ALABAMA (ATOFMS, PALMS, SPLAT, LAAPTOF, LAMPAS…)-technique can only yield one particle type, while AMS will report the 4 quantities. Both of them are correct, though.

Organic aerosol reported from AMS measurement is frequently further resolved by PMF analysis, and the resulting factors are typically HOA, COA, BBOA, OOA (sometimes divided into LV-OOA and SV-OOA). In the ALABAMA data we detected similar particle types: engine exhaust (similar to HOA), biomass burning particles (BBOA), and even cooking emissions (although we have the interference with sea salt in this special case). The OOA (oxygenated organic aerosol) component is typically interpreted as aged organic aerosol by the AMS user community (to which we also belong). Thus, OOA-dominated particles will fall into the "aged material" type in the single particle analysis. As no anions were available during the Jungfraujoch measurements, nitrate and sulfate could not directly be detected. $NH_4^+$ (m/z 18), however, is an indicator for ammonium nitrate or ammonium sulfate. The particle assigned to the particle type "aged material" contain both the organic marked fragments and the NH4 markers, such that we assume internal mixture of secondary inorganic and organic particles.

We also have to note here that single particle laser ablation mass spectrometry is not a new technique. There have been numerous publications since about 1998 that use automatic clustering algorithms and marker peak search for the assignment of particle spectra to certain particle types and report the results as pie charts (or other form of graphic visualization). We therefore think that a detailed and quantitative discussion of instrument sensitivity towards different species is beyond the scope of this manuscript. We use the same ablation laser at the ATOFMS instrument or the SPASS instrument (a 266 nm Nd:YAG), so we can refer the reader to earlier publications of the Prather group (e.g. Pratt and Prather, 2010; Pratt et al.,

2009; .), other ATOFMS users (e.g., Sierau et al., 2014; Kamphus et al., 2010), and SPASS users (Erdmann et al., 2005; Hinz et al., 2006).

We added a short paragraph to section 2.2:

Typically, the assignment of mass spectra to a certain particle type relies on the most abundant marker peaks. Therefore, smaller species that are abundant on many or even all particle types might go unnoticed. Thus is most likely the case for secondary inorganics (sulfate, nitrate) and secondary, oxygenated organics, which may add a coating to mineral dust particles, but the particle signal is still dominated by the mineral dust signatures. Thus it has to be kept in mind that a particle type called "mineral" should be read as "mineral-dominated". The only exception we made here is the particle type "lead-containing" where we explicitly state that lead does not represent the whole particle composition. Such a classification of particles is very common in the single particle mass spectrometry literature. The ALABAMA uses a 266 nm ND:YAG laser for particle ablation and ionization, thus we expect the assignment of mass spectra to be similar to other instruments using the same laser wavelength (ATOFMS, SPASS) of which many results on the abundance of particle types similar to our classification have been reported (Pastor et al., 2003; Erdmann et al., 2005; Hinz et al., 2006; Pratt et al., 2009; Kamphus et al., 2010; Pratt and Prather, 2010; Sierau et al., 2014).

12-
As the authors note, the present data do not allow ice nucleation rates for different particle types to be determined. Therefore please change "particles have good ice nucleating ability" to "particles were observed within ice crystal residuals" on p12,26 and please change "The high ice nucleation ability of etc etc could be confirmed" to "The presence of INP from etc etc was observed". This avoids overstating the results, which are nonetheless valuable and interesting.
The authors may also improve their manuscript by comparing the relative fractions of each particle type they have observed to previous studies of IPR or IN composition. That is, by comparing to the Cziczo citation given later in this review and the various citations already present in the manuscript. Although ice nucleation rates cannot be determined, a quantitative comparison in another dimension can nevertheless be added.

We changed the text as suggested.

We included a more detailed comparison of our observed particle type in IPR with those reported by Kamphus et al from the same site and also by Cziczo et al., 2013. However, it has to emphasized that Cziczo 2103 discussed cirrus clouds, where the formation processes, especially temperatures, are very different to mixed phase clouds.

Added text:

The general trend of this finding agrees with the only result so far in the literature on single particle mass spectrometric analysis of IPR from mixed phase clouds (Kamphus et al., 2010): Using two single particle mass spectrometers, they report from one instrument (SPLAT) that 57% of all IPR were mineral particles or mixtures of minerals with sulfate, organics, and nitrate. The other instrument (ATOFMS) reported that these two particles types represent a much higher fraction (78%) of all IPR, plus additionally 8% metallic particles. However, these data sets are based on smaller numbers of particle than our study (ATOFMS 152 particles, SPLAT 355 particles), such that here variations of air mass origin and meteorological conditions can be the main reason for such differences and none of these data sets can be regarded as representative for mixed phase clouds at the Jungfraujoch in general. A recent paper by Cziczo et al. (2013) summarized their analyses of ice crystals on various field

studies. Although formation of cirrus clouds occurs under different conditions that ice formation in mixed phases clouds, it is interesting to compare these results as well. These data clearly show that mineral dust is the most dominant heterogeneous ice nucleus in all cirrus encounters, but that under homogeneous freezing conditions the upper tropospheric background aerosol particles (sulfate/organic/nitrate) as well as biomass burning particles are detected in the cirrus IPR.

Minor:
- Percentages of particle types are frequently reported, but it is not always clear what the reference is (% of all IPR vs % relative to all of that particle type?). Please give a universal definition in Section 2. Please also add uncertainties to each reported %.

We added the definition of the percentage to section 2

"We report the absolute number of particles of the particle type in a certain time period and the percentage of these particles to the total particle population (i.e, the sum of all particle types) measured during the same time period."

We added uncertainties (as explained above) to all values.

Section 3.2.1. Some particle types are given no discussion at all (engine and PAH) while others are given extensive discussion (industrial and lead). If there is some reason why engine and PAH particles were not further discussed, please make a brief note for the reader.

As already mentioned above, we added a discussion on engine exhaust particles and PAH-containing particles on in section 3.2.1:

The particle types that were assigned to the type "engine exhaust" also show $C_n$-fragmentation $(C_1^+ - C_5^+$, m/z 12 … 60) but can be distinguished by the peak at m/z 40 ($Ca^+$) which was observed in the reference mass spectra but also previously by other researchers (Trimborn et al., 2002; Vogt et al., 2003; Shields et al., 2007).

PAH containing particle were identified through the corresponding reference spectra and marker peaks from the laboratory studies, namely 50/51 ($C_4H_{2/3}^+$), 63 ($C_5H_3^+$), 77 ($C_6H_5^+$), and 91 ($C_7H_7^+$). Even though cigarette particles (before inhalation) contain these markers as well, our reference spectra indicate that these two particle types can be distinguished because cigarette smoke additionally contains a $C_n$ pattern (m/z $12 - 36$).

- p3,21 This description of LDI (laser desorption/ionization) incorrectly implies two-step vaporization and ionization. The laser does not "vaporize a fraction of created gas molecules" since ionization can occur during desorption. Better would be "the pulsed laser fires and vaporizes/ionizes the particle partly or completely."

Agreed. We changed the text accordingly.

- p3,25 what refractive index was assumed for particle sizing?

The Grimm 1.129 is calibrated with PSL particles, refractive index = 1.60. We added this information in section 2.1.

- p4,33 cooking is not combustion, so change "directly produced by combustion" to "directly sampled from the source"

Changed.

- p4,33 I believe "supernatant" is the scientific term for "washing water"

Agreed, we used the scientific term but kept "washing water" in parentheses since it was used in previous publications (Augustin et al., 2013; Augustin-Bauditz et al., 2016).

p5,3 when using the word majority, please give a number

The number of spectra representing the majority differs between particle types. The supplement now shows the spectra from the laboratory measurement. It was found that one reference particle type produced several types of spectra which were separated using the clustering algorithm described above. The supplement lists the main clusters with the number of spectra in each cluster.

p5,29 does the manual switching mean that there is some bias in the results? Was there always a cloud free period measured at the end of the in cloud periods? Please clarify.

"Manual switching" means that no automated valve was used that would be controlled by a cloud droplet sensor. When a cloud was present, we sampled though the ice-CVI, when no cloud was present, we sampled through the aerosol inlet. Sampling though the CVI under non-cloud condition corresponds to a zero measurements because no particles enter the CVI.

We changed the text to: "Under cloud conditions the ALABAMA sampled through the Ice-CVI, whereas under cloud-free condition it was switched manually to the total aerosol inlet."

p6,14, what is meant by "ambient temperatures below 0 C"? Either ambient, or below 0 C.

We rephrased this sentence to make it more clear:

The impaction plates of the pre-impactor are not actively cooled but adopt ambient temperature which must be below 0°C to allow for mixed phase clouds to exist.

p7,18 please define "PAH fragmentation" including citations. Can "PAH" be recognized separately from "aromatic"?

The fragmentation pattern of PAHs was measured during our laboratory studies. We included the mass spectra in the supplement.
We rephrased the sentence: "The particles from smoke after inhalation do not show the characteristic marker peaks that were observed for PAH particles in the laboratory study."

Unfortunately no other aromatics were measured in the laboratory, thus we cannot make a statement on this issue.

p7,20 please give the physical reason why only sodium or potassium would be observed.

This is due to the non-uniform laser ablation and ionization process. Sodium and potassium are present in pollen and biomass burning particles. Apparently (as the laboratory data show) it occurs that only these two ions are produced upon laser ablation/ionization.

We reformulated the sentence:

"It was observed that some rare fragmentation patterns from pollen and biomass burning particles show similarities within the cation spectra (only one sodium (m/z 23) and potassium (m/z 39) peak). This is most likely due to the non-uniform laser ablation and ionization process, leading to production of only those two ions."

p9,30 please cite a paper for this point

This statement was a tentative explanation of the finding that sometimes only a single potassium peak is observed in the cation mass spectrum. While there are references confirming that laser ablation is very sensitive to potassium (Gross et al., 2000; Silva and Prather, 2000), we can't find a reference for the speculation that other ions are suppressed. Thus, we rephrased the sentence:

It must be taken into account that the detection of potassium in laser ablation mass spectrometry is very efficient due to its low ionization efficiency (Gross et al., 2000; Silver and Prather, 2000), such that only a small amount of potassium in a particle results in a large ion signal.

p9,34 I don't understand the motivation behind this sentence. The message seems to be that a similar number fraction of biomass burning particles in the aerosol and in the IPR is an unexpected result. Why would it be unexpected?

We reformulated the whole paragraph, as it was indeed hard to understand. It reads now:

The IPR ensemble contains particles assigned to engine exhaust but no particles assigned to biomass burning. This finding is surprising because both particle types have been detected in IPR before: biomass burning by Kamphus et al. (2010), Twohy et al. (2010), Pratt et al. (2011) and Prenni et al. (2012), and engine exhaust by Kamphus et al. (2010) and Corbin et al. (2012). Both engine exhaust and biomass burning particle contain organic material and the mass spectra show partly similar peaks, thus it is unclear which specific property of these particle types enables their ice nucleation ability. The third particle type associated with combustion (PAH/soot) is found in both populations in the same percentage (12%). The source of these particles can also be biomass burning or engine exhaust (as well as other combustion processes), but the finding that the percentage in both populations is equal suggests that the PAH/soot components are not significantly influencing the ice nucleation capability.

Fig 2 and 6. Please add uncertainties for all values. Please sort the table by either out-of-cloud or IPR values. Please add a column to the table "%IPR / %out-of-cloud" to estimate the relevance enhancement of each category, including uncertainties.

We added the uncertainties that were estimated as explained above, and also added the suggested column giving the enhancement. However, we prefer to keep both pie charts and the

table in the order as it is now. The particle types are ordered from biological/natural to anthropogenic/industrial, followed by the unassigned, descriptive types "K dominated" and "Na + F" and "others".

-The following articles should be cited in this manuscript: Jaenicke, Abundance of Cellular Material and Proteins in the Atmosphere, Science, 308, 5718, pp 73, 2005. Cziczo et al., Clarifying the Dominant Sources and Mechanisms of Cirrus Cloud Formation, Science, 340, 6138, pp 1320-1324.

We included citations to these articles.

Very minor comments:

p1,20, unclear wording. try, "the outcome of these laboratory studies was particle type specific marker peaks for each investigated particle type."

Changed to: "The outcome of these laboratory studies was characteristic marker peaks for each investigated particle type. These marker peaks were applied to the field data."

p2,5 starting from "therefore" is a repetition of the previous statement which takes some thinking to realize.

We deleted the second part of the sentence.

p2,22 please also discuss Cziczo et al., Science 2013.

We have included a reference to Cziczo et al., 2013.

p2,28 "first results" to me is synonymous with "preliminary results". Perhaps better is "Inspection of the data set showed"

Changed

p10,35-37. "does not represent the real distribution" is more clearly phrased as "is not corrected for sampling and detection efficiency"

Changed to:

"It must be emphasized here that the ALABAMA size distribution is not corrected for sampling and detection efficiency, the latter being optimal around 400 nm."

p11,27 with temperature and relative humidity, the wet-bulb temperature is already given?

Yes, but not straightforward (Stull, 2011), thus we preferred to look at the wet-bulb temperature directly as it was supplied by MeteoSwiss.

[revised manuscript text omitted]

---

## Author Comment (AC2) · 7 Oct 2016

acp-2016-365

**Schmidt et al., On-line single particle analysis of ice particle residuals from mountain-top mixed-phase clouds using laboratory derived particle type assignment**

Reply to Reviewer #2

Reviewer comments and questions are printed in this font type.

>Our replies are printed like this.

>Changes to the manuscript text are printed in blue.

The manuscript by Schmidt et al. presents ambient and laboratory single particle mass spectrometer data to identify chemical components in aerosols and ice residuals on Jungfraujoch and to draw conclusions on the ice nucleation ability of different aerosol particle components. Given the importance of ice nucleation processes for precipitation and the climate, and the many uncertainties related to these processes, such studies are needed. The combination of laboratory experiments and ambient measurements makes this study especially useful for the single particle mass spectrometer community. I therefore recommend publication of this well-written manuscript in ACP after the following comments have been addressed:

>We thank the reviewer for the positive rating of our manuscript

General comments

The limitations and uncertainties of single particle mass spectrometry and the respective data should be addressed in more detail.

There are several aspects:
- More care should be given as how one describes the quantities. It should be made clear, e. g. in the introduction, that "a larger relative amount" cannot be interpreted as "more aerosol mass", but can only refer to "a larger number of particles" assigned to a certain category.

>Agreed. This is a very important point for laser ablation single mass spectrometry. We replaced "relative amount" by "number fraction" in the abstract, in section 3.2.2 and 3.2.4, and in the summary.

- The presentation of marker mass fragments and reference spectra from different particle types is highly desirable for the single particle mass spectrometer community. For the spectra to be useful for other groups/instrument types however, information on uncertainties needs to be given. One of the issues of laser ablation single particle mass spectrometers is the (weak) repeatability of spectra/measurements; depending on particle size, chemical composition/morphology, and on the placement of the particle in the laser beam, the amount of ablated/ionized material can vary significantly. According to p. 4, l. 36, you were not size-selecting particles for your laboratory tests, which might have led to larger spectrum-to-spectrum differences. Showing e. g. standard deviations of the averaged spectra would give an idea on the uncertainty and variability of the marker spectra.

We have now added the reference mass spectra to the supplement. We show the most frequent spectra types (clusters) for each substance. This illustrates the variability of the mass spectra that are obtained from one particle type.

We consider standard deviations of the averaged mass spectra not as an ideal estimate of the uncertainty, because the abundance of a marker peak is the main criterion, not its height in the mass spectrum.

- Related to the above – how were the marker peaks identified? Given the overlap of marker peaks between different particle types, it seems to me that presenting the data as marker spectra would almost make more sense. How important are the individual markers for identification as opposed to the whole spectrum? In Table 3 you give the ratio of the number of spectra containing marker peaks (all marker peaks? just some?) to all spectra of a particular particle type. A better way to present uncertainties would be the standard deviation of the avergaed spectra (see above). The ratios in Table 3 are fairly low, I thus expect a relatively high scatter of spectra per particle type.

As already mentioned above, we included the reference spectra in the supplement. The scatter of spectra per particle type is illustrated by showing the main spectra types (clusters).

Regarding the overlap of marker peaks: It is the combination of the characteristic marker peaks that is used to assign a mass spectrum to a certain particle type. In cases where there is a complete overlap of all marker peaks between reference spectra, we can't distinguish these substances. As said above, the standard deviation of the averaged mass spectra are not the best estimate of uncertainty, because the abundance of the peak is more important that its height.

- More information should be given on the choice of clustering algorithm and the uncertainties of its outcome. There are a lot of references given to Roth (2014, 2016), but the main points should be conveyed to the reader in the manuscript (also in Table 1). A general result I infer from the ambient measurements is that secondary (here in the sense of formed from a chemical reaction/in combustion) components (both organic and inorganic) or particles containing a lot of secondary (organic) components (e. g. biomass burning) are less effective ice nuclei than primary organic particles such as e. g. biological particles. In this simple categorization, however, it is hard to place the engine exhaust particles. It would be helpful if the engine exhaust and PAH particle types were discussed in the manuscript (which is not the case now).

We have included a general explanation of the clustering and of the chosen algorithm to section 2.2 (as requested also by Reviewer #1). We also added a discussion on PAH and engine exhaust particles.

Changes to the text describing the clustering (Section 2.2):

Basically, a clustering algorithm tries to find the optimum number of clusters (i.e. groups of mass spectra) that represent the particle population by their average mass spectrum. By nature of the aerosol particle diversity and the non-uniform ionization in laser ablation ionization, it can't be expected that all particles contained in a cluster equal the average cluster spectrum (Hinz et al., 1999). Rather, each spectrum is assigned to that cluster where the distance metric (in our case one minus Pearson's correlation coefficient r) of the single particle spectrum and the averaged cluster spectrum reaches a minimum. The fuzzy c-means algorithm differs from the k-means in that way that it accounts for the possibility that one particle may belong to two

(or more) clusters by using membership coefficients, whereas the k-means assigns each spectrum strictly to that cluster where correlation with the averaged spectrum is highest.

Here we applied the fuzzy c-means algorithm because sensitivity tests conducted in the framework of a PhD thesis (Roth, 2014) with laboratory-generated particle of known composition and number have shown that the fuzzy c-means better separates the particle types and suffers less from false assignments. Also, various parameters that influence the clustering result where tested by Roth (2014), resulting in a "best choice" that was applied here as well: All mass spectra were normalized to reduce the influence of total signal intensity, and all m/z peaks were taken to the power of 0.5 to reduce the influence of the non-uniform laser ablation ionization, thereby increasing the influence of smaller peaks and decreasing that of larger signals. The "fuzzifier", a weighting exponent used for the calculation of the membership coefficients (Bezdek, 1982; Roth et al., 2016) was set to 1.2. A high number of start clusters was chosen to assure that also rare spectra types are considered in the data evaluation (for instance for the out-of-cloud data set from the JFJ campaign 2013 a number of 200 cluster was chosen; for a known particle composition as sampled during the laboratory studies a number of 10 to 50 cluster was chosen depending on the number of spectra). The start clusters were chosen randomly from the total particle population, under the condition that the correlation coefficient (r) between two randomly picked start spectra is less than 0.7. The procedure leading from the clustering algorithm to a certain number of particle types, as illustrated in Fig. 1, was as follows: After mass calibration, the spectra were clustered using the fuzzy c-means algorithm, yielding a certain number of clusters. The resulting cluster number can be lower than the chosen number of start clusters. If this is the case, the number of start clusters was sufficiently high not to suppress rare spectra types. Each cluster includes a certain number ($\geq 1$) of mass spectra based on the calculated membership and distance. From all mass spectra in a cluster an average spectrum is calculated which is used for the identification of the particle type represented by each cluster. All mass spectra which did not fulfill the distance criterion ($1 - r \leq 0.3$) compared to any of the clusters were sorted in the cluster "others". The averaged spectra of each cluster were manually examined with respect to the presence of the marker peaks derived from the reference mass spectra (Section 3.1) and assigned to a certain particle type. The "others" cluster was processed again using the fuzzy c-means algorithm, but with reduced constraints and again the resulting clusters were manually examined and, if possible, assigned to particle types. At the end all clusters of the same particle type were merged, whereas clusters that could not be assigned to a certain particle type were added to the cluster "others".

Changes to text with respect to PAH and engine exhaust particles (section 3.2.1):

The particle types that were assigned to the type "engine exhaust" also show $C_n$-fragmentation ($C_1^+ - C_5^+$, m/z 12 … 60) but can be distinguished by the peak at m/z 40 ($Ca^+$) which was observed in the reference mass spectra but also previously by other researchers (Vogt et al., 2003).

PAH containing particle were identified through the corresponding reference spectra and marker peaks from the laboratory studies, namely 50/51 ($C_4H_{2/3}^+$), 63 ($C_5H_3^+$), 77 ($C_6H_5^+$), and 91 ($C_7H_7^+$). Even though cigarette particles (before inhalation) contain these markers as well, our reference spectra indicate that these two particle types can be distinguished because cigarette smoke additionally contains a $C_n$ pattern (m/z 12 – 36).

I am confused about the similarity of sea salt and cooking emissions fragmentation pattern. How were the sea salt particles produced, what kind of cooking emissionswere investigated? The variety in cooking activities is incredibly large (food cooked?what method? fuel used? etc.), and without further information a "cooking spectrum"is not very meaningful.

> The similarity in the fragmentation pattern of sea salt and cooking emissions only holds for the cations. Unfortunately, only cations were available from the Jungfraujoch field data (as explained in section 2.4.1). The cation spectra (with m/z 23, 39, 46, 81, 83, 139) occur both in sea salt and in barbecue emissions, and we assume that they are from salt contained in the spicing of the meat and the cheese.
>
> In Table 2 (now Table 1) we noted that we sampled the particles directly from a barbecue (charcoal) in outside air. We used sausages, steaks and cheese. The marker peaks listed in Table 3 (now Table2) were found in all three spectra types. We will add this information to Section 2.3 and will refer to this particle type as "cooking/barbecue emissions" throughout the whole text in the revised version.
>
> The sea salt particles were produced atomizing a solution of commercially (Sigma Aldrich) available sea salt.

Soil dust and minerals are categorized as "natural" aerosol, in accordance with classifications in literature. However, soil dust aerosol concentrations can directly be influenced by anthropogenic activities (e. g. farming, mining, forestry).These are all important factors, especially when looking into/modelling anthropogenic influences on climate via aerosol-cloud interactions. I am fully aware that soil dust source apportionment lies outside the scope of this paper, but I suggest leaving out the word "natural" and potentially add a sentence on this issue.

> We agree with the reviewer and have left out the word "natural" when describing the dust particles. However, biological particles can be attributed to natural sources. We therefore refer to "primary and/or natural sources", and added a brief explanation of the different source types at the beginning of section 3.2.2:
>
> In comparison to the out-of-cloud aerosol the IPR ensemble shows a higher number fraction of particles from primary and/or natural sources. We attribute biological and sea salt to primary natural sources, whereas soil dust and minerals emissions can directly be influenced by anthropogenic activities and can therefore not be regarded as purely natural. Biomass burning particles generated from forest fires are not primary particles, but can be related to natural sources as well.

Specific comments

P. 3 - 4, l. 37 – 2: Please give the reason for choosing the fuzzy c-means algorithm over the other two possibilities.

> As mentioned above, we included a general explanation of the clustering and of the chosen algorithm to section 2.2.:
>
> Basically, a clustering algorithm tries to find the optimum number of clusters (i.e. groups of mass spectra) that represent the particle population by their average mass spectrum. By nature of the aerosol particle diversity and the non-uniform ionization in laser ablation ionization, it

can't be expected that all particles contained in a cluster equal the average cluster spectrum (Hinz et al., 1999). Rather, each spectrum is assigned to that cluster where the distance metric (in our case one minus Pearson's correlation coefficient r) of the single particle spectrum and the averaged cluster spectrum reaches a minimum. The fuzzy c-means algorithm differs from the k-means in that way that it accounts for the possibility that one particle may belong to two (or more) clusters by using membership coefficients, whereas the k-means assigns each spectrum strictly to that cluster where correlation with the averaged spectrum is highest.

Here we applied the fuzzy c-means algorithm because sensitivity tests conducted in the framework of a PhD thesis (Roth, 2014) with laboratory-generated particle of known composition and number have shown that the fuzzy c-means better separates the particle types and suffers less from false assignments.

P. 5, l. . 3 – 9: See general comment on uncertainties and limitations of marker/spectra identification.

We added the description of the error estimation and included the error ranges in the tables and throughout the text.

Changes to text (section 2.2)

The uncertainties reported along with these numbers were estimated by manual inspection of a subset of the data, as described in Roth et al. (2016). The assignment of a certain cluster to a particle type is based on the presence of the reference marker peaks in the averaged cluster mass spectrum. Upon inspection of all mass spectra in one cluster it may occur that the marker peaks (or not all of the marker peaks) are not present in an individual mass spectrum. Such a mass spectrum has nevertheless been correctly (from a mathematical point of view) assigned to the cluster by the algorithm, because the overall correlation of the mass spectrum with the cluster average is sufficiently high (r > 0.7). This can especially occur in cases when many other peaks are similar, as is often observed for organic particles.

For the error estimation, such particle mass spectra were regarded as "uncertain assignments". The percentage of such uncertainly assigned mass spectra was regarded as the relative error. Of the out-of-cloud data set we inspected two clusters, one assigned to biological particles (338 particles) and one assigned to biomass burning aerosol (473 particles). It turned out that 52 of the 338 inspected "biological" mass spectra (15%), and 48 of the 473 inspected "biomass burning" mass spectra (10%) had to be considered as uncertain. Thus, we conservatively estimated the relative error to be about 15% and generalized this error for whole out-of-cloud data set.

For the IPR data set, where the absolute numbers of particles are much lower, it was possible to do a more detailed inspection of the clusters: We inspected one cluster assigned to biological particles, where we found that 28 out of 76 were uncertain (37%), and one cluster of the "PAH/soot" particle type, where 9 out of 23 spectra were uncertain (40%). Those particle types containing only a small number of particles ("industrial metals", "Na + K", "aged material") were completely inspected manually, yielding uncertainties for the "industrial metals" of 14%, of the "Na + K" type of 0% (no uncertain particles), and of the "aged material" type of 44%. Thus, we estimated the relative error (from uncertain particle type assignment) of the IPR population to be 40% with the exception of the industrial metals (14%) and the "Na + K" type.

These error estimates are conservative upper limits for the error range, because the reference laboratory measurements have shown that, e.g., not all biological particles contain the characteristic marker peaks. It may therefore well be that mass spectra that are similar to the cluster average spectrum of a "biological particle" type are really biological particles, even if

they do not contain the marker peaks. The uncertainty inferred from manual inspection was combined with the Poisson counting statistics error (by error propagation) for each particle type.

P. 5, l. 25 – 28: Was the inlet heated to 20_C to prevent condensation? Did you perform any assessment of the influence of the heating on the chemical composition (e. g. evaporation of semi-volatile material)? What was the residence time in the inlet? The evaporation of semi-volatile material could be especially important for SIA and SOA. Were parts of the ICE-CVI heated as well? Please add more information on sampling/inlet conditions.

Weingartner et al (1999) describe the inlet and give two reasons for heating the inlet:

"Therefore the main inlet at the Sphinx observatory was designed to sample the interstitial aerosol as well as the activated droplets. The inlet consists of a heated and insulated vertical stainless-steel tube (length: 200 cm; diameter: 6 cm) and a heated snow-hood. The temperature of the sampled air was measured 15 cm downstream of the inlet entrance and was electronically regulated to +20°C. As already mentioned, heating was necessary (1) to dry essentially all activated droplets as early as possible to reduce transmission losses and (2) to prevent riming of the inlet system during harsh conditions, especially in winter."

We can therefore refer to the Weingartner-Paper for details of the inlet and it is not necessary to repeat this information in our manuscript.

The Ice-CVI was not heated (see Mertes et al., 2007). We did not assess the influence of the heating of the total inlet on the chemical composition, and we can't think of a method that allows for doing this. But, if heating of the total led to a loss of SIA and SOA, it would result in an underestimation of particles of the type "aged material" in the out-of-cloud aerosol. This would not affect the finding that the number fraction of "aged material"-particles is much smaller in IPR than in out-of-cloud aerosol.

P. 5, l.34: What about the transmission of particles with diameters larger than 500 nm? A relatively large fraction of the particles measured at Jungfraujoch was larger than 500 nm, according to Figure 3.

The transmission decreased from 99% to 95 between 500 and 1000 nm, thus sampling line losses are not an issue for the interpretation of the ALABAMA data. We added this information to section 2.4.1:

For both sampling lines the transmission efficiency was about 99 % for particle sizes between 200 nm and 500 nm and decreased to 95 % for particle sizes up to 1000 nm. The upper 50 %-cut-off of the Ice-CVI was at about 4900 nm and for the total inlet about 3300 nm.

P. 7, l. 7 – 8: Many of the marker mass fragments of the biological particles (especially bacteria and pollen) have negative marker mass fragments. How were they identified in ambient air where you only had positive spectra?

In the positive spectra we used the peaks at m/z 47 ($PO^+$) and at m/z 58 ($C_3H_8N^+$) and 59 ($C_3H_9N^+$), the latter two indicating trimethylamine. We named this particle type "biological/amine".

P. 9, l. 31 – 38: See general comment above. A more thorough discussion on the properties and uncertainties of engine exhaust and PAH spectra in comparison with biomass burning spectra might shed some light on their differences in ice nucleation behavior.

We added a description on how PAH and engine exhaust were identified:

The particle types that were assigned to the type "engine exhaust" also show $C_n$-fragmentation ($C_1^+ - C_5^+$, m/z 12 … 60) but can be distinguished by the peak at m/z 40 ($Ca^+$) which was observed in the reference mass spectra but also previously by other researchers (Vogt et al., 2003).

PAH containing particle were identified through the corresponding reference spectra and marker peaks from the laboratory studies, namely 50/51 ($C_4H_{2/3}^+$), 63 ($C_5H_3^+$), 77 ($C_6H_5^+$), and 91 ($C_7H_7^+$). Even though cigarette particles (before inhalation) contain these markers as well, our reference spectra indicate that these two particle types can be distinguished because cigarette smoke additionally contains a $C_n$ pattern (m/z 12 – 36).

P.10, l. 34 – p. 11, l. 3: Please elaborate further on the comparison and discrepancy of OPC and ALABAMA size distributions. The size distributions shown in Figure 3 are completely different and basically do not allow to draw any conclusions. Are there no artefacts of the Sky-OPC?

We chose to include the size distribution measured with the OPC because the size distribution measured with the ALABAMA is not representative as a total number size distribution. The black lines in a) and c) show the number of analyzed particles in each size bin. From this it can be seen that the ALABAMA detection efficiency decreases for particles smaller than 400 nm. However, the relative number fraction of particle types in each class is not affected by this, as long as there are sufficient particles per size bin. It is therefore possible to apply the relative composition as a function of size to the size distribution measured by the OPC which we regard here as a "realistic" distribution. There are certainly artefacts to an optical particle counter, but this discussion is beyond the scope of our paper.

A second reason why we chose to show the size distribution measured by the OPC is the marked difference between the IPR and the out-of-cloud aerosol particles: The slope of the distribution is much steeper for the out-of-cloud aerosol, showing that larger particles are relatively more abundant in IPR than in the out-of-cloud aerosol.

We restructured the paragraph describing the size distributions and added more explanation.

P.11, l. 22 – 34: Whereas the meteorological conditions were similar for the two periods, they do no coincide in termns of time of day, which however can have large influences on (anthropogenic) emission patterns (e. g. engine exhaust, cooking: : :). Please take this into account in your data interpretations.

It was not possible to find two events with similar meteorological conditions, air mass origin and time of day. We added the following text:

Further, it has to be noted that both sampling periods differ in their sampling length and their time of day: The IPR sampling period lasted almost from midnight to noon, while the corresponding out-of-cloud sampling period lasted only 72 minutes in the afternoon. This may lead to different aerosol particle population due to different emission patterns of anthropogenic particles like engine exhaust, cooking etc. However, it was not possible to find two sampling periods having the same time of day and similar meteorological conditions and air mass

origins. Also, the IPR sampling period could not be shortened because a sufficient number of particle needs to be sampled for a meaningful analysis.

P. 12, l. 27: This "conclusion" is a rather minor finding of your study (or put in different words, a potential reason for differences in composition between activated and non-activated aerosol particles that can be ruled out). This sentence should not be at the end of the results section.

We deleted this sentence here because this conclusion (scavenging does not play role) has already been discussed in section 3.2.2

P. 13, l. 1 – 28: Again, in your summary, please a add a few sentences on uncertainties and the limitation of the method concerning single particle marker spectra identification and particle type detection/identification in ambient air.

We added the following text at the beginning of the summary:

Uncertainties of the method arise from the finding that not all particle mass spectra from one particle type display the characteristic marker peaks. This is a result of the non-uniform ionization process of laser ablation particle mass spectrometry. For some particle types (pollen, sea salt, cooking/barbecue emissions) the fraction of mass spectra showing characteristic marker peaks was high (60 – 84 %), whereas for mineral particles (desert dust, soil dust etc.) the percentage of reference mass spectra with specific markers peaks was markedly lower (20 – 37 %). The resulting particle assignment to particle types can't be corrected for this effect, but it is likely that this is the cause for the large fraction of "unknown" particles (17 – 19 %).

Technical comments

P. 1, l. 19 – 21: "As outcome: : :" – weird sentence structure, rephrase

We rephrased to: "The outcome of these laboratory studies was characteristic marker peaks for each investigated particle type. These marker peaks were applied to the field data."

P. 1, l. 32: "and" instead of "an"

This was meant to be "an". We added a comma behind "clouds", maybe this helps:

"Depending on their chemical and microphysical properties aerosol particles have a strong impact on the solar radiation budget, an influence on the life-time of clouds, and hence also on precipitation."

P. 4, l. 37: Phrase structure; should read "Coating experiments were also…"

Changed.

P.7, l. 4 – 5: Weird sentence structure

We do not see why this sentence is unclear.

P. 9, l. 5: Dot at the end of the sentence is missing.

Corrected.

P. 12, l. 15: Sentence structure: Should read "It has to be noted further..."

We deleted this paragraph with respect to a comment of reviewer #1.

P. 12, l. 27: conclusion

Corrected.

**References**

Bezdek, J. C., Ehrlich, R., and Full, W.: FCM: The fuzzy c-means clustering algorithm, Computers & Geosciences, 10, 191-203, 1984.

Hinz, K.-P., Greweling, M., Drews, F., and Spengler, B.: Data Processing in On-line Laser Mass Spectrometry of Inorganic, Organic, or Biological Airborne Particles, American Society for Mass Spectrometry, 10, 648-660, 1999.

Mertes, S., Verheggen, B., Walter, S., Connolly, P., Ebert, M., Schneider, J., Bower, K. N., Cozic, J., Weinbruch, S., Baltensperger, U., and Weingartner, E.: Counterflow Virtual Impactor Based Collection of Small Ice Particles in Mixed-Phase Clouds for the Physico-Chemical Characterization of Tropospheric Ice Nuclei: Sampler Description and First Case Study, Aerosol Science and Technology, 41, 848-864, 2007.

Roth, A.: Untersuchungen von Aerosolpartikeln und Wolkenresidualpartikeln mittels Einzelpartikel-Massenspektrometrie und optischen Methoden, PhD thesis, University of Mainz, Germany, urn:nbn:de:hebis:77-37770, 2014.

Roth, A., Schneider, J., Klimach, T., Mertes, S., van Pinxteren, D., Herrmann, H., and Borrmann, S.: Aerosol properties, source identification, and cloud processing in orographic clouds measured by single particle mass spectrometry on a Central European mountain site during HCCT-2010, Atmospheric Chemistry and Physics 15, 24419-24472, 2016.

Vogt, R., Kirchner, U., Scheer, V., Hinz, K. P., Trimborn, A., and Spengler, B.: Identification of diesel exhaust particles at an Autobahn, urban and rural location using single-particle mass spectrometry, Journal of Aerosol Science, 34, 319-337, doi: http://dx.doi.org/10.1016/S0021-8502(02)00179-9, 2003.

Weingartner, E., Nyeki, S., and Baltensperger, U.: Seasonal and diurnal variation of aerosol size distributions (10 < D < 750 nm) at a high-alpine site (Jungfraujoch 3580 m asl), Journal of Geophysical Research-Atmospheres, 39 104, 26809-26820, 1999.

---

## Editor Comment (EC1) · A. K. Bertram (Editor) · 18 Oct 2016

Editor comment:

After the authors posted their responses to the referee comments on MS No.: acp-2016-365, I received a "follow-up comment" from one of the referees of this manuscript. To expedite the review process, I am posting this "follow-up comment" as an editor comment. The authors of MS No.: acp-2016-365 should also consider this "follow-up comment" when revising their manuscript. Below is the initial comment from the reviewer, the response by the authors, and the "follow-up comment".

[Figure]

Initial comment from the reviewer:

3- In Section 2.3 when discussing the lab spectra it is stated "only those mass spectra that represented the majority of the different fragmentation patterns were considered". What % defines majority? Moreover, this % should be used to define a correction factor (with corresponding uncertainty propagated into the result). If only 60% of particles were measured for salt but 90% for soot, then the reported IPR numbers should be scaled up by 1/0.6 and 1/0.9 respectively.

Response by the authors:

Correction of the reported IPR numbers can't be done in this way. First, we might end up with more particles that were actually measured. Second, it is only possible to assign marker ions of atmospheric particles to those reference spectra containing these marker ions. Speculating about spectra not containing the markers would only increase the uncertainty.

"follow-up comment" from the reviewer:

I'm not quite convinced by the arguments in this response. It would be appropriate to end up with more particles than were measured, if a correction for missed particles was applied. There are 2 ways the ALABAMA may miss particles. The obvious way is if the instrument only obtains a signal for, say, 10% of all particles (90% "missed"). Then a correction factor 1/0.1 should be applied. The second way is if the instrument obtains a full mass spectra (with marker ions) for 10 of 100 biomass burning particles and only a partial mass spectra (eg no signals except potassium) for 30 of 100 biomass burning particles. We can suppose that their remaining 60 are missed particles. In the second case, the correct number to be reported for the biomass burning class is 10*1/0.1. There is ambiguity about what to do with the 30 partial mass spectra: they could be put into a category "other" but careful thought would be needed before reporting total estimates for number concentration.

**[ACPD]{.underline}**

Interactive
comment

It is essential to include such a correction factor if the goal is to compare the abundances of different particle types. If a correction factor is not applied, it must be proven that the conclusions are unchanged by its omission – it is only valid to omit the correction if it would be similar for all particle types, in analogy to weighting or not weighting a linear regression.

Therefore I still think that the manuscript should include a report (discussion/table/graph) of the fraction of laboratory samples which included the marker ions and which did not. If this fraction was very different between classes, the comparative statistics in the abstract (13% dust, 3% aged particles in IPR) would be incorrect.

There is another subtlety which I would like to state but do not expect the authors to address. A biomass burning particle missing certain markers may possibly be classified as aged, which would mean a second-order correction of overlapping particle classes could be made (this is just an example, and not based on the authors' marker ions). Without this correction, particle type numbers could indeed be overestimated, but I imagine this would be pushing the data analysis past what is reasonable.

---

## Author Comment (AC3) · 1 Nov 2016

Reply to editor's comment (follow-up comment by Reviewer #1)

Initial comment from the reviewer:

3- In Section 2.3 when discussing the lab spectra it is stated "only those mass spectra that represented the majority of the different fragmentation patterns were considered". What % defines majority? Moreover, this % should be used to define a correction factor (with corresponding uncertainty propagated into the result). If only 60% of particles were measured for salt but 90% for soot, then the reported IPR numbers should be

scaled up by 1/0.6 and 1/0.9 respectively.

Response by the authors:

Correction of the reported IPR numbers can't be done in this way. First, we might end up with more particles that were actually measured. Second, it is only possible to assign marker ions of atmospheric particles to those reference spectra containing these marker ions. Speculating about spectra not containing the markers would only increase the uncertainty.

"follow-up comment" from the reviewer:

I'm not quite convinced by the arguments in this response. It would be appropriate to end up with more particles than were measured, if a correction for missed particles was applied. There are 2 ways the ALABAMA may miss particles. The obvious way is if the instrument only obtains a signal for, say, 10% of all particles (90% "missed"). Then a correction factor 1/0.1 should be applied. The second way is if the instrument obtains a full mass spectra (with marker ions) for 10 of 100 biomass burning particles and only a partial mass spectra (eg no signals except potassium) for 30 of 100 biomass burning particles. We can suppose that their remaining 60 are missed particles. In the second case, the correct number to be reported for the biomass burning class is 10*1/0.1. There is ambiguity about what to do with the 30 partial mass spectra: they could be put into a category "other" but careful thought would be needed before reporting total estimates for number concentration.

It is essential to include such a correction factor if the goal is to compare the abundances of different particle types. If a correction factor is not applied, it must be proven that the conclusions are unchanged by its omission – it is only valid to omit the correction if it would be similar for all particle types, in analogy to weighting or not weighting a linear regression.

Therefore I still think that the manuscript should include a report (discussion/table/graph) of the fraction of laboratory samples which included the marker ions and which did not. If this fraction was very different between classes, the comparative statistics in the abstract (13% dust, 3% aged particles in IPR) would be incorrect. There is another subtlety which I would like to state but do not expect the authors to address. A biomass burning particle missing certain markers may possibly be classified as aged, which would mean a second-order correction of overlapping particle classes could be made (this is just an example, and not based on the authors' marker ions). Without this correction, particle type numbers could indeed be overestimated, but I imagine this would be pushing the data analysis past what is reasonable.

——————————————————————————————————

Reply by Johannes Schneider (on behalf of all co-authors):

We considered the follow-up comment very carefully, but we still think that such a correction is not possible. We list our arguments here:

1) "missed particles": It would be necessary to determine a size-resolved detection efficiency for all reference particles. We did not size-select the test particles, but parallel measurements with an optical particle sizer are available. However, some uncertainty would result from this, because of the necessary conversion of the different equivalent diameters.

2) Assuming such size-dependent correction factors are available, a major obstacle is that parameters inferred from pure, laboratory-generated particles or directly from the source sampled particles can not be transferred to aged, processed particles found in the ambient atmosphere. The shape of freshly emitted soot or combustion particles will change through atmospheric processing from very non-spherical to more spherical, thereby improving the focusing properties of the particles in the aerodynamic lens. Thus, the detection efficiency will improve by processing. Also, non-spherical dust particles can be coated by secondary organic or inorganic material, leading to a more spherical shape. Matrix effects, altering the ionizations process of internally mixed

particles compared to the pure laboratory particles are also possible.

3) Particles that are ionized but do not show the marker peak spectra: Here the same argument as above holds: We would apply correction factors inferred for pure laboratory particles to aged and processed atmospheric particles.

4) Particles that are ionized but that do not show the marker peaks would in the current analysis either assigned erroneously to another particle type or to the group "others". Since it is not known where these falsely assigned particle end up, we can't correct for this error, too. Please note also that the cluster "others" is by far not large enough to account for all possible particle not showing the marker peaks. Additionally, although we investigated a large number of reference particles, there will by a large number of possible atmospheric particle types that we did not investigate and which may be included in the "others" group. Thus, subtracting particle from the "others" cluster is not a valid option.

Summarizing, we think that trying to correct for all these effects would lead to unacceptably high uncertainties and should not be done, neither for ALABAMA not for any other single particle mass spectrometer, and has to our knowledge not been done by other groups.

In contrast, we prefer to understand the relative abundances and absolute numbers of particles as "number of particles identified by the ALABAMA" and not as "number of particles in the ambient atmosphere". We will clarify this in the text and in the figure captions. We will also add a discussion of the points above to the manuscript.

Nevertheless, comparison between the relative abundances of particle types in out-of-cloud aerosol and ice residuals (which is the focus if this manuscript) is possible, because, to the best of our knowledge, the above effects affect both particle populations in the same way.